# Tmem65 is critical for the structure and function of the intercalated discs in mouse hearts

The intercalated disc (ICD) is a unique membrane structure that is indispensable to normal heart function, yet its structural organization is not completely understood. Previously, we showed that the ICD-bound transmembrane protein 65 (Tmem65) was required for connexin43 (Cx43) localization and function in cultured mouse neonatal cardiomyocytes. Here, we investigate the functional and cellular effects of Tmem65 reductions on the myocardium in a mouse model by injecting CD1 mouse pups (3–7 days after birth) with recombinant adeno-associated virus 9 (rAAV9) harboring Tmem65 shRNA, which reduces Tmem65 expression by 90% in mouse ventricles compared to scrambled shRNA injection. Tmem65 knockdown (KD) results in increased mortality which is accompanied by eccentric hypertrophic cardiomyopathy within 3 weeks of injection and progression to dilated cardiomyopathy with severe cardiac fibrosis by 7 weeks post-injection. Tmem65 KD hearts display depressed hemodynamics as measured echocardiographically as well as slowed conduction in optical recording accompanied by prolonged PR intervals and QRS duration in electrocardiograms. Immunoprecipitation and super-resolution microscopy demonstrate a physical interaction between Tmem65 and sodium channel β subunit (β1) in mouse hearts and this interaction appears to be required for both the establishment of perinexal nanodomain structure and the localization of both voltage-gated sodium channel 1.5 (NaV1.5) and Cx43 to ICDs. Despite the loss of NaV1.5 at ICDs, whole-cell patch clamp electrophysiology did not reveal reductions in Na+ currents but did show reduced Ca2+ and K+ currents in Tmem65 KD cardiomyocytes in comparison to control cells. We conclude that disrupting Tmem65 function results in impaired ICD structure, abnormal cardiac electrophysiology, and ultimately cardiomyopathy.

The ICD is a unique cellular structure in cardiomyocytes and is functionally responsible for coordinating muscle contraction. Genetic mutations that impair ICD protein function often cause arrhythmogenic, hypertrophic, or dilated cardiomyopathies in human patients and animals[1]. Four types of junctional complexes have been identified in the ICD, including adherens, gap, ephaptic, and desmosomes junctions. These junctions are functionally intertwined and provide chemoelectrical communication and mechanotransduction between adjacent cardiomyocytes[2–4]. Both the mechanisms behind the assembly of these junctions and the mode of inter-junctional interaction,

✉e-mail: Allen.Teng@utoronto.ca; Anthony.Gramolini@utoronto.ca

however, remain mostly an enigma. A better understanding of ICD biology will provide a greater insight to its functional adaptation during cardiac pathophysiology.

Desmosomes/Adherens junctions provide contact points for cell cytoskeletons at the ICD plicate. While desmosomes are a nodal point for intermediate filament desmin, adherens junctions are nodal points for filamentous actins. In some regions of ICDs, both junctions can physically intertwine, forming *area composita*[5]. Each desmosome complex begins with homo- or heterodimerization of transmembrane proteins desmocollin 2 (DSC2) and desmoglein 2 (DSG2) that interact with armadillo proteins plakoglobin (PG) and plakophillin 2 (PKP2)[5,6]. Desmoplakin (DSP) bridges the armadillo proteins and desmin. Similarly, adherens junctions begin with homodimerized transmembrane N-Cadherin (NCAD) that interacts with β-catenin/PG complexes which bind to actin filaments via α-catenin[5,6]. Many of these proteins are vital to ICD integrity, as dysfunction in these proteins results in heart failure in transgenic animals and humans[7].

The main function for gap and ephaptic junctions is the transjunctional ion flux at the ICD. Gap junction protein Cx43 forms hexameric complexes on adjacent cardiomyocytes for unilateral ion flux between cells[8,9], while ephaptic junctional protein voltage-gated sodium channel (NaV1.5) mediates fast influx of $Na^+$ ions from the perinexal extracellular space[4]. NaV1.5 is also involved in mechanotransduction through ankyrin-G (Ank-G), which is a cytoskeletal scaffold that interacts with Cx43 and PKP2[10]. Ank-G knockout in mouse hearts is associated with internalized Cx43, impaired desmosomes, and aberrant ICDs structures[11]. Conversely, Cx43 trafficking and localization to the ICD depends on desmosomes[12], possibly through a specialized ICD region called the perinexus. Both NaV1.5 and sodium channel beta subunit 1 (β1) are enriched in the perinexus[3,4,13,14], but the underlying mechanism for the trafficking and localization of these proteins to the perinexus are mostly unknown. These questions are of great interest to perinexus biology, especially given a recent study by Jiang et al. reported no evidence of direct interaction between NaV1.5 and β1[15]. Identification of β1-interacting proteins at the ICD may begin elucidating the underpinning biology.

Transmembrane protein 65 (Tmem65) is a cardiac-enriched protein[16]. It interacts with Cx43 and DSP at the ICD in adult mouse hearts. Loss of Tmem65 function is associated with disrupted conduction velocity in cultured mouse neonatal cardiomyocytes in a Cx43-dependent fashion[16]. In this study, we investigated the role of cardiac Tmem65 in vivo. Our findings show that Tmem65 and β1 physically interacted in mouse hearts and this interaction was necessary for the localization of NaV1.5 and Cx43 to the ICD. shRNA-mediated Tmem65 knockdown led to irregular staining patterns of perinexal proteins, altered $Ca^{2+}$ and $K^+$ currents, impaired cardiac conduction, and ICD structural defects in both cellular and histological levels. Tmem65 KD mouse hearts ultimately developed dilated cardiomyopathy, severe fibrosis, and congestive heart failure. Our study has demonstrated Tmem65 as a new perinexal protein that is critical for the assembly of perinexal complexes and its function in the ICD.

## Results

### Tmem65 KD leads to dilated cardiomyopathy, fibrosis, and increased mortality

We previously showed that Tmem65 silencing led to a reduction and internalization of Cx43, resulting in disrupted conduction in cultured mouse neonatal cardiomyocytes[16]. To examine the impact of Tmem65 KD on cardiac function, we created a mouse model of reduced Tmem65 by injecting mouse newborns (3–7 days post birth) with rAAV9-Tmem65 shRNA (or rAAV9-scrambled shRNA as control) intraperitoneally (Fig. 1a). We found that intraperitoneal injections (both rAAV9-Tmem65 and rAAV9-scrambled shRNA) resulted in increased deaths in Tmem65 KD mice. In particular, Kaplan–Meier survival plot revealed that most male mice survive the first 21 days post injection

(Fig. 1b), but showed health declines characterized by lethargy, disheveled hairs, and breathlessness. All the male Tmem65 KD mice died after 60 days post injection, while most female mice (11 of 13 total, 85%) survived until weeks 6–7 even though they also showed similar signs of illness (Supplemental Movie 2) as described for males, with evidence of congestive heart failure (CHF). In comparison, control mice were physically active and well-groomed (Supplemental Movie 1).

Echocardiographic measurements (Fig. 1c) of female mice, surviving to 6–7 week pot-injection, revealed evidence of heart failure as characterized by severely reduced ($P < 0.01$) cardiac hemodynamic parameters in Tmem65 KD mice (Table 1; $n = 6$/group) with an 81% reduction in ejection fraction ($14.83 \pm 3.92\%$ in Tmem65 KD mice; $76.75 \pm 3.05\%$ in control mice) and an 86% reduction in fraction shortening ($5.33 \pm 1.50\%$ in Tmem65 KD mice; $39.10 \pm 2.69\%$ in control mice). In addition, the anterior and posterior wall thickness of the left ventricle in Tmem65 KD mice decreased ($P < 0.01$) when compared to those of controls (Fig. 1c and Table 1), implying dilated cardiomyopathy. Depressed cardiac hemodynamics of Tmem65 KD hearts was also confirmed by Doppler flow velocity spectrum that measured the aortic flow velocity-time integral (VTI) at the aortic orifice. Figure 1d shows a consistent and significant decreased ($P < 0.01$) VTI in Tmem65 KD mice ($1.91 \pm 0.19$ cm) compared to control mice ($4.83 \pm 0.55$ cm). In addition, aortic blood flow was irregular with velocities that were clearly altered, or near-zero in some cases, across several contractions (Fig. 1d, yellow arrows). ECG changes were evident in Tmem65 KD compared to control (Fig. 1d), suggesting conduction abnormalities (ECG, Fig. 1d).

The hearts of Tmem65 KD mice were also grossly enlarged in histological sections 7 weeks after injection compared to control hearts (Fig. 1e, f) with the frequent appearance of thrombi in the left atrium, which was never seen in controls (Fig. 1e, black arrow). Further inspection of H&E-stained tissues revealed patches of eosin-negative lesions that were enriched with hematoxylin-positive nuclei in Tmem65 KD hearts (Fig. 1g). We postulated that these regions were the result of either fibroblast or immune cell infiltration. Immunohistochemistry of macrophage marker F4/80 was comparable between groups, while Masson trichrome staining showed profound increases ($P < 0.01$) of cardiac fibrosis in Tmem65 KD hearts (Fig. 1h, i, $91.24 \pm 5.57$ A.U. in Tmem65 KD heart; $8.75 \pm 8.36$ A.U. in control hearts). These findings demonstrate that a long-term (6–7 weeks) suppression of Tmem65 expression leads to dilated cardiomyopathy phenotype with severe cardiac fibrosis.

Collectively, the results above demonstrate clear evidence of dilated cardiomyopathy by 6–7 weeks after intraperitoneal injection of Tmem65 shRNA. This was associated, as expected, with decreased Tmem65 expression after only 3 weeks post Tmem65 KD with 85% reductions in Tmem65 mRNA levels ($14.4 \pm 2.5\%$ remaining) and reduction in both Tmem65 protein isoforms ($7.1 \pm 0.7\%$ remained for 25 kDa Tmem65; $26.8 \pm 7.4\%$ remained for 17 kDa Tmem65) compared to control hearts (Fig. 1j, k, Supplementary Fig. 1).

### Tmem65 KD leads to eccentric hypertrophic cardiomyopathy and slowed cardiac conduction by 3 weeks post injection

Since male mice began dying about 3 weeks following Tmem65 KD, when Tmem65 expression was profoundly reduced, we sought to characterize the mice at the 3-week time point in the hopes of uncovering the basis for the devasting impact of Tmem65 reduction on heart function and mouse longevity. We found that the Tmem65 mice appeared physically indistinguishable from control mice. Nevertheless, isolated hearts (ventricles and left atria) from Tmem65 KD mice were morphologically larger than control hearts in histological sections (Fig. 2a) which was associated with increased heart weight-to-tibia length ratios ($11.21 \pm 0.45$ mg/mm in Tmem65 KD versus $7.45 \pm 0.32$ mg/mm in control) (Fig. 2b). Moreover, ventricular cross sections showed dilated lumens of both ventricles in Tmem65 KD hearts than control hearts, while the ventricular wall thickness in both

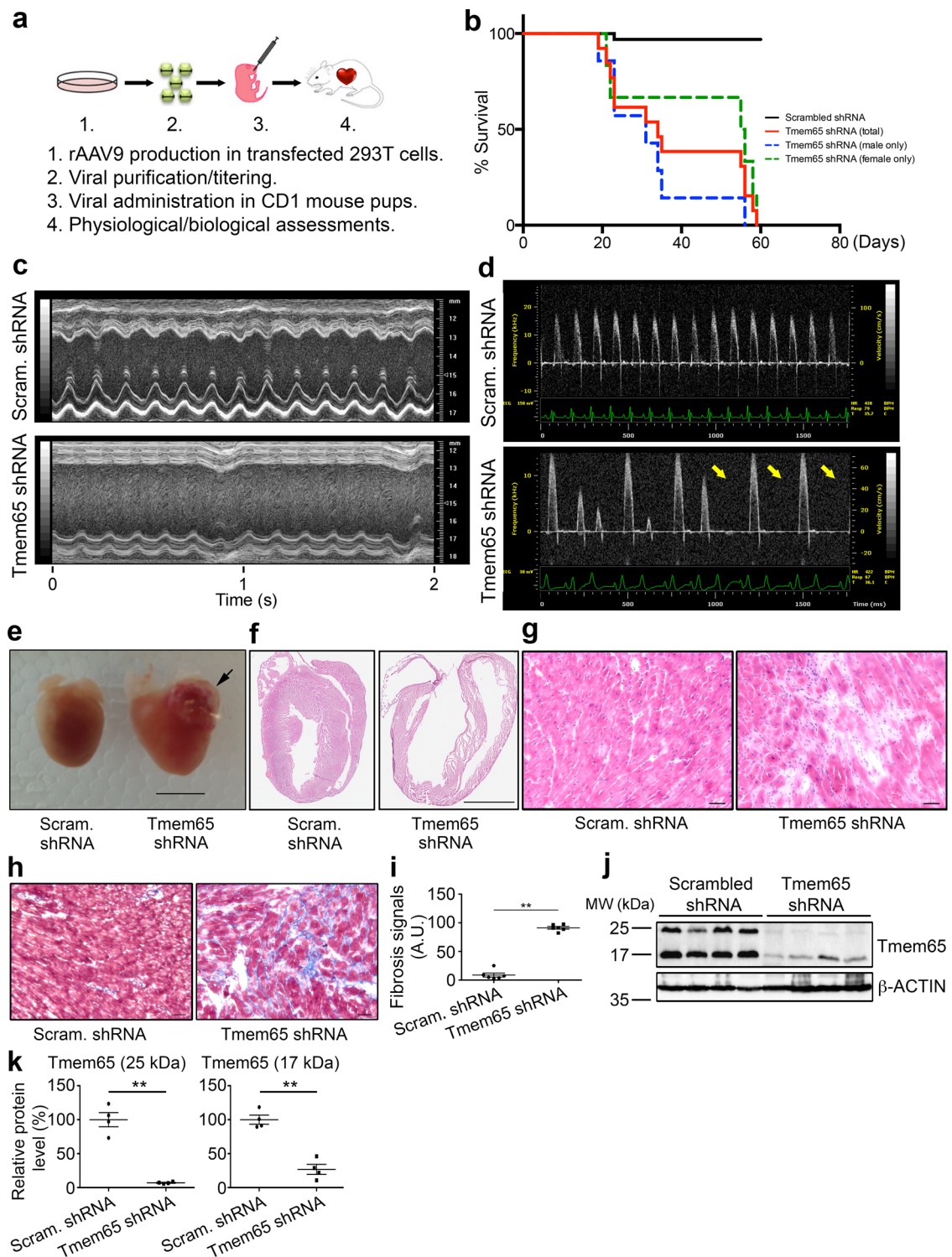

groups was comparable (Fig. 2c), suggesting eccentric hypertrophic cardiomyopathy. Phalloidin (F-actin) staining of isolated adult ventricular myocytes showed myofibrillar disorganization in Tmem65 KD myocytes (Fig. 2d), but sarcomeric F-actins remained aligned in control myocytes. Dimensional measurements of isolated myocytes revealed significant increases ($P < 0.01$) in both the length ($114.2 \pm 2.5\,\mu m$ in Tmem65 KD myocytes; $102.4 \pm 1.3\,\mu m$ in control myocytes) and width ($38.6 \pm 1.0\,\mu m$ in Tmem65 KD myocytes; $30.2 \pm 0.6\,\mu m$ in control myocytes) in Tmem65 KD mice (Fig. 2e). Finally, we tested several biomarkers for cardiac hypertrophy and immunoblots (Fig. 2f), which showed increased ($P < 0.01$) protein levels for α-ACTININ ($153 \pm 16\%$

Tmem65 KD hearts; $100 \pm 13\%$ control hearts), cardiac troponin T (cTnT; $178 \pm 20\%$ Tmem65 KD hearts; $100 \pm 16\%$ control hearts), four-and-half LIM domain 1 (FHL1; $788 \pm 153\%$ Tmem65 KD hearts; $100 \pm 12\%$ control hearts), β-myosin heavy chain (MYH7; $187 \pm 37\%$ Tmem65 KD hearts; $100 \pm 16\%$ control hearts), and phosphorylated extracellular signal-regulate kinases 1/2 (p-ERK1/2, $159 \pm 21\%$ Tmem65 KD hearts; $100 \pm 22\%$ control hearts). qPCR showed the mRNA of natriuretic peptide A (Nppa) and B (Nppb) was significantly elevated ($P < 0.01$) in Tmem65 KD hearts (Fig. 2g. Nppa, $14.96 \pm 0.57$ Tmem65 KD hearts, $1.10 \pm 0.52$ control hearts; Nppb, $4.58 \pm 0.46$ Tmem65 KD hearts; $0.75 \pm 0.22$ control hearts). Together, our results show that a short-

**Fig. 1 | Tmem65 KD leads to dilated cardiomyopathy, fibrosis, and increased mortality. a** Schematic illustration for establishing a Tmem65 KD mouse model. **b** Kaplan–Meier survival curve showing all Tmem65 KD mice (red line, $n = 30$ total mice; blue dotted line, $n = 16$ males; green dotted line, $n = 14$ female) died within 7 weeks, while scrambled control mice (blue line, $n = 29$) remained healthy. Greater than 50% of Tmem65 KD mice died 3 weeks post injection. Experiments were performed in mice of both sexes. **c** M-mode recordings of mouse left ventricles showed depressed cardiac contractility in a Tmem65 KD mouse heart compared to a scrambled control heart. **d** Pulsed wave Doppler measurements at aortic orifice measuring blood velocity in a Tmem65 KD mouse heart and control mouse heart. The velocity of blood flow was reduced, or flow waveform was irregular or missing (yellow arrows) in Tmem65 KD mouse hearts. Echocardiography was performed in 6 female mice per group. **e** Morphology of Tmem65 KD and controls hearts. Tmem65 KD hearts were grossly larger compared to control hearts, including both atria and ventricles. Blood clots were always found in the left atrium (black arrow). Scale bar = 5 mm. **f** H&E staining of atria and ventricles of Tmem65 KD and control hearts. **g** Higher magnification of H&E-stained hearts showing many eosin-negative lesions, haematoxylin-positive nuclei. **h** Masson trichrome staining showing collagen deposits (blue color). Scale bar = 50 μm. **i** Quantification of cardiac fibrosis of cardiac tissues stained with Masson Trichrome dye. A nearly tenfold increase in fibrosis in Tmem65 KD hearts compared to control hearts. \*\*$P < 0.01$. Histopathology was performed in 6 female mouse hearts. **j** Immunoblots demonstrating changes of Tmem65 protein levels in Tmem65 KD hearts. β-actin was included as a loading control. **k** Densitometry quantification of Tmem65 levels in **j**. Both long (25 kDa) and short (17 kDa) forms of Tmem65 were compared to the control samples. \*\*$P < 0.01$, $n = 4$. All statistical analyses were performed by one-way ANOVA with Tukey's post-hoc test. Data were expressed as mean ± standard error of the means.

term (3 weeks) Tmem65 silencing is associated with eccentric hypertrophic cardiomyopathy.

Our previous results showed that Tmem65 KD led to aberrant ICD junctions. Since ICD is the hub for electrical and mechanical cell–cell communication between cardiomyocytes[4,17–19], we assessed indirectly cardiac conduction. Optical mapping experiments in isolated hearts (Fig. 2h) revealed that, compared to control hearts ($n = 6$, 4 male and 2 female), Tmem65 KD hearts ($n = 6$, 4 female and 2 male) displayed reduced ($P < 0.05$) conduction velocity both when hearts beat in sinus rhythm (45.6 ± 1.3 cm/s Tmem65 KD; 56.7 ± 3.9 cm/s control hearts) and when paced at 11 Hz from the right ventricular free wall (24.0 ± 1.1 cm/s Tmem65 KD; 35.9 ± 1.0 cm/s control hearts) (Fig. 2i, Table 2). Consistent with altered cardiac conduction, we found that the ECGs (Supplementary Fig. 2, Table 3) of conscious Tmem65 KD mice had increased ($P < 0.05$) PR intervals (28.89 ± 3.85 ms Tmem65 KD hearts; 22.70 ± 1.85 ms control hearts) as well as prolonged QRS durations (16.35 ± 0.36 ms Tmem65 KD hearts; 10.47 ± 0.42 ms control hearts).

## Tmem65 silencing was associated with ICD defects in mouse hearts

Since our previous studies in cultured cardiomyocytes revealed reduced Cx43 at the ICDs[16], we speculated that the slowed conduction

might be related to altered ICD structure and function. Consistent with our previous results, we observed decreases ($P < 0.05$) in Cx43 protein (33.4 ± 12.8%) and mRNA (30.3 ± 5.7%) in the Tmem65 KD group 3 weeks post injection (Fig. 3a, Supplementary Fig. 1a). Moreover, immunofluorescence revealed that the Pearson's correlation coefficient (PCC) for co-localization between Cx43 and NCAD, an intercalated disc marker, was reduced by about 50% in Tmem65 KD hearts (i.e. 0.72 ± 0.05 in control versus 0.33 ± 0.07 in Tmem65 KD hearts), demonstrating increased internalization of Cx43 in Tmem65 KD hearts within 3 weeks of Tmem65 KD (Fig. 3b, c, Supplementary Fig. 1b). Despite these changes in co-localization, NCAD protein levels were comparable between control and Tmem65 KD hearts. To investigate the location of internalized Cx43 proteins, we also examined the PCC between Cx43 and the mitochondrial marker Cox4 and found a significant ($P < 0.05$) increase in Cx43/Cox4 co-localization in Tmem65 KD hearts (0.67 ± 0.05 in Tmem65 KD hearts versus 0.57 ± 0.04 in control, Fig. 3d, e). Together, these findings demonstrate the establishment of an in vivo mouse model for the loss of Tmem65 expression and that silencing Tmem65 in mouse hearts recapitulates aberrant Cx43 protein levels and distribution as previously observed in cultured mouse neonatal cardiomyocytes.

Altered ICD structure was also observed in transmission electron microscopy studies, which showed marked structural changes of ICD morphology in Tmem65 KD hearts, characterized by a sharp 'zig-zag' or elongation pattern in the Tmem65 KD myocardium compared to control hearts (Fig. 3f, black arrows). These changes were quantified[20], to establish trends ($p = 0.062$) towards ICD lengthening (1.93 ± 0.7 μm in Tmem65 hearts versus 1.36 ± 0.4 μm in control hearts) as well as reduced ($P < 0.01$) ICD curvature/ICD length ratios (1.49 ± 0.47 in Tmem65 KD hearts versus 2.67 ± 0.40 in control hearts) as summarized in Fig. 5g. These findings suggest that the changes in ICD structure and Cx43 localization to the ICDs may underlie the slowed cardiac conduction seen in the Tmem65 KD mice.

The structural studies above establish that Tmem65 KD leads to a loss in ICD structure. Since intercellular adhesion at ICDs are dominated by three structures (desmosomes, adherens junctions and fascia adherens), we examined the structure and localization of the desmosome protein PKP2 which is important to desmosome stability[2,10] and desmin, which creates multiple mechanical connections with F-actin in the sarcomere, mitochondria and the sarcoplasmic reticulum[21,22]. Consistent with an important role of Tmem65 in desmosome structure, we observed that Tmem65 KD reduced ($p < 0.05$) co-localization of PKP2 or DSG2 with NCAD (PCC was 0.47 ± 0.09 in control hearts versus 0.24 ± 0.06 in Tmem65 KD hearts), as summarized in Fig. 3h, i. Moreover, further evidence for the loss of desmosome integrity is provided by the reduced sarcomeric desmin levels (Fig. 3j, white arrows) combined with an aberrant longitudinal desmin staining pattern (Fig. 3j, yellow arrow) in Tmem65 KD hearts which was associated with a "split" appearance of the sarcomere myofibrils on either side of ICDs (15 split myofibrils/36 total myofibrils in 5 electron micrographs

## Table 1 | Morphological and functional assessments of the left ventricles by echocardiography in scrambled control and Tmem65 KD mice

| | Scrambled shRNA ($n = 6$) | Tmem65 shRNA ($n = 6$) |
|---|---|---|
| AWes (mm) | 1.36 ± 0.06 | 0.6 ± 0.10\*\* |
| ESD (mm) | 2.31 ± 0.16 | 4.30 ± 0.14\*\* |
| PWes (mm) | 1.13 ± 0.08 | 0.69 ± 0.08\*\* |
| AWed (mm) | 0.85 ± 0.14 | 0.53 ± 0.09\*\* |
| EDD (mm) | 3.78 ± 0.14 | 4.54 ± 0.13\*\* |
| PWed (mm) | 0.71 ± 0.01 | 0.62 ± 0.07\* |
| EF (%) | 76.75 ± 3.05 | 14.83 ± 3.92\*\* |
| FS (%) | 39.10 ± 2.69 | 5.33 ± 1.50\*\* |
| Stroke volume (μl) | 47.64 ± 5.25 | 14.58 ± 1.72\*\* |
| Heart rate (beats/min) | 419.33 ± 29.36 | 455.83 ± 12.25 |
| Cardiac output (ml/min) | 19.27 ± 1.46 | 6.63 ± 0.52\*\* |

Echocardiography was performed on 6 female mice in each group 6 weeks post viral injection. *AWes* anterior wall thickness at end-systole, *ESD* end-systolic diameter of left ventricular chamber, *AWed* anterior wall thickness at end-diastole, *EDD* end-diastole diameter of left ventricular chamber, *PWed* posterior wall thickness at end-diastole, *PWes* posterior wall thickness at end-systole, *EF* ejection fraction, *FS* fraction shortening, *mm* millimeter, *μl* microliter, *ml* milliliter, *min* minute.
\**p* < 0.05, \*\**p* < 0.01.

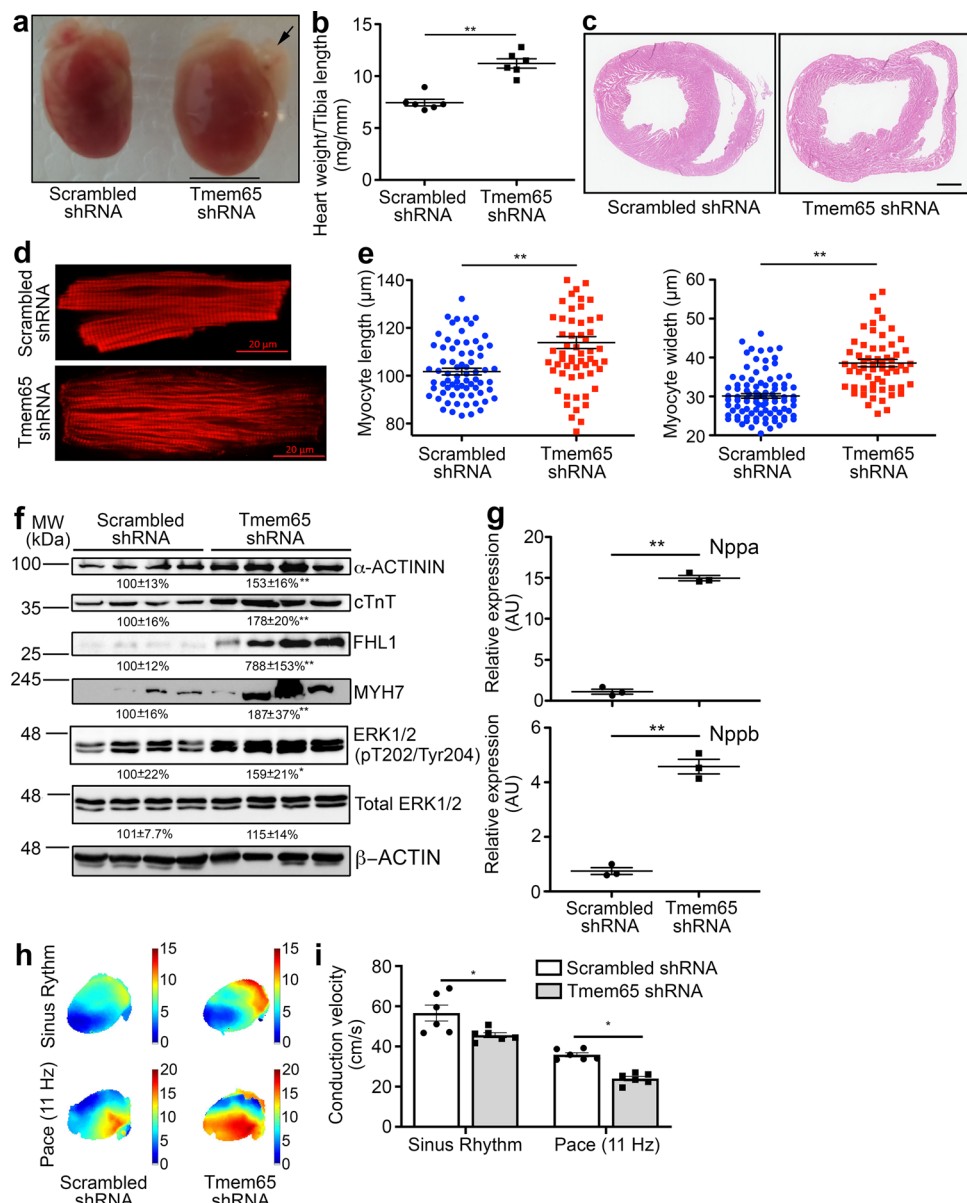

**Fig. 2 | Tmem65 KD leads to eccentric hypertrophic cardiomyopathy and slowed cardiac conduction 3 weeks post injection. a** Morphology of Tmem65 KD and controls hearts. Tmem65 KD hearts were qualitatively larger than control hearts, especially the left atrium (black arrow). (scale bar = 5 mm). **b** Measurement of heart weight-to-tibia length ratio in Tmem65 KD and controls hearts. **$P < 0.01$, $n = 6$ per group. Statistical analyses were performed by one-way ANOVA with Tukey's post-hoc test. Data were expressed as mean ± standard error of the means. **c** H&E staining of cardiac transverse sections. Scale bar = 500 μm. **d** Phalloidin (F-actin) staining of adult mouse cardiomyocytes isolated from Tmem65 KD or control hearts. Scale bar = 20 μm. **e** Quantifications of cell dimensions of isolated adult mouse cardiomyocytes. Both lengths (left panel) and width (right panel) significantly increased in Tmem65 KD cardiomyocytes. **$p < 0.01$, $n > 90$ cells. Three mice per group. Statistical analyses were performed by one-way ANOVA with Tukey's post-hoc test. Data were expressed as mean ± standard error of the means. **f** Immunoblots of multiple hypertrophic markers including α-ACTININ, cTnT, FHL1, MYH7, and phosphorylated ERK1/2 with densitometry quantifications. **$P < 0.01$;

*$P < 0.05$. Statistical analyses were performed by one-way ANOVA with Tukey's post-hoc test. Data were expressed as mean ± standard error of the means. **g** qPCR showing increased transcript levels of natriuretic peptide A and B (**$P < 0.01$) in Tmem65 KD hearts. Experiments were performed in more than 3 mice of both sexes. Statistical analyses were performed by one-way ANOVA with Tukey's post-hoc test. Data were expressed as mean ± standard error of the means. **h** Representative images of cardiac optical mapping. Transfer of voltage-sensitive dye di-4-ANEPPS was monitored in control and Tmem65 KD hearts at the Sinus rhythm (top images) or paced at 11 Hz (bottom images). **i** The conduction velocity of the right ventricular free wall of Tmem65 KD hearts ($n = 6$ hearts, 4 females and 2 males) was significantly lower than that of control hearts at the Sinus rhythm or paced at 11 Hz (*$P < 0.05$). All cells were isolated from male mouse hearts. For optical mapping, data were expressed as mean ± SEM. Differences between scrambled shRNA and Tmem65 shRNA groups at sinus rhythm and during pacing were assessed with a two-way ANOVA with Sidak's multiple comparison test. Data were presented as mean ± standard deviation.

from 4 Tmem65 KD hearts versus 0 split myofibrils/32 total myofibrils in 4 electron micrographs from 3 control hearts). Similar observations were found in the isolated Tmem65 KD adult mouse cardiomyocyte (Fig. 2d). Together, these studies suggest that Tmem65 is needed for the maintenance of desmosome junction integrity and thereby Cx43 co-localization. These findings are also consistent with our previous

study showing co-localization and interactions between desmoplakin and Tmem65[16].

Morphological changes in mitochondria have often been associated with ICD defects[21,23,24], but we found no clear distinction in mitochondrial morphology between 2 groups. However, mitochondria were loosely scattered around the ICD in Tmem65 KD hearts, while

**Table 2 | Optical mapping assessments of right ventricular free walls of control and Tmem65 KD hearts 3 weeks post viral injection**

|  | Scrambled shRNA | | Tmem65 shRNA | |
|---|---|---|---|---|
|  | Sinus rhythm | Pace (11 Hz) | Sinus rhythm | Pace (11 Hz) |
| APD25 (ms) | 13.1 ± 0.3 | 17.7 ± 1.8 | 13.6 ± 0.6 | 21.8 ± 1.0* |
| APD50 (ms) | 23.8 ± 1.1 | 26.1 ± 1.4 | 24.5 ± 1.6 | 30.8 ± 1.0* |
| APD80 (ms) | 48.3 ± 1.3 | 39.7 ± 3.3 | 47.8 ± 2.3 | 42.1 ± 1.1 |
| Time-to-peak (ms) | 6.8 ± 0.4 | 9.8 ± 0.2 | 7.4 ± 0.8 | 11.8 ± 0.4* |

Four male and 2 female mice in Tmem65 group; 4 female and 2 male mice in Tmem65 group.
*APD* action potential duration.
*p < 0.05.

**Table 3 | Cardiac electrical conduction assessment by electrocardiography of Tmem65 KD or scrambled control mice**

|  | Scrambled shRNA (n = 10) | Tmem65 shRNA (n = 8) |
|---|---|---|
| RR (ms) | 72.45 ± 2.08 | 88.43 ± 1.53** |
| PR (ms) | 22.7 ± 1.85 | 28.89 ± 3.85** |
| QRS (ms) | 10.47 ± 0.42 | 16.35 ± 0.36** |
| QTc (ms) | 45.19 ± 1.79 | 54.30 ± 1.94** |
| ST (ms) | 28.37 ± 1.72 | 34.91 ± 1.66** |
| HR (bpm) | 415 ± 10 | 347 ± 16** |

ECG was performed on 10 control male mice and 8 Tmem65 KD male mice.
*ms* milliseconds, *bpm* beats per minute.
**p < 0.01.

they were densely backed at the ICD of control hearts (Fig. 3f, Supplementary Fig. 3). Indeed, in Tmem65 KD hearts, significantly less ($P < 0.01$) mitochondria were in contact with neighboring mitochondria at the ICD (0.93 ± 0.12 contacting mitochondria) when compared to that of control hearts (2.38 ± 0.15 contacting mitochondria) (Supplementary Fig. 4). Finally, mitochondria clustering at non-ICD regions were comparable between both groups (Supplementary Fig. 3, left and central panels).

**Tmem65 silencing also leads to losses of Nav1.5 channels at ICDs**
Our results demonstrate that a short-term (3 weeks) reduction in Tmem65 protein levels is associated with aberrant ICD junctions in mouse hearts. Previous studies have shown that cardiac Na$^+$ channels (i.e., NaV1.5) are also found, along with β1 subunits, at the ICD regions[4]. Similar to Cx43, immunoblots demonstrated that NaV1.5 expression was reduced ($P < 0.01$) in Tmem65 KD hearts (Fig. 4a, 62.8 ± 0.1% in Tmem65 KD hearts; 100 ± 0.4% in control hearts), while β1 protein levels were decreased ($P < 0.01$) by ~50% in Tmem65 KD hearts (Fig. 4a., 52.2 ± 0.2% in Tmem65 KD hearts; 100 ± 0.6% in control hearts). Immunofluorescence in Tmem65 KD hearts further showed abnormal NaV1.5 and β1 staining patterns that did not appear to span uniformly the ICD (Fig. 4b), with NaV1.5 being absent from ICDs of Tmem65 KD hearts (Fig. 4b, top right panel), without affecting sarcolemmal NaV1.5 staining (Supplementary Fig. 5). On the other hand, immunofluorescence of β1 showed either dispersed puncta (Fig. 4b, bottom right panel, white arrows) or aggregates in Tmem65 KD hearts (Fig. 4b, bottom right panel, yellow arrows), but remained largely at ICDs in control hearts (Fig. 4b, bottom left panel). A previous study proposed that β1 could be needed for NaV1.5 and NCAD interaction via an unknown mechanism[14]. We thus speculated that loss of β1 function could affect NaV1.5 docking and retention at the ICD in Tmem65 KD hearts. Co-immunoprecipitation assays with anti-NCAD antibody showed decreased NaV1.5 signals in Tmem65 KD hearts compared to control cardiac samples (Fig. 4c, top panels).

These results demonstrate the abrogated NaV1.5 at the ICD in Tmem65 KD hearts via an unknown mechanism.

To assess the functional consequences of changes in NaV1.5 expression at the ICD on reductions in conduction velocity with Tmem65 KD, we measured sodium membrane currents ($I_{Na}$) using voltage-clamp measurements of isolated cardiomyocytes (Fig. 4d). Somewhat unexpectedly, peak $I_{Na}$ current densities were comparable ($P = 0.680$) between the control and Tmem65 KD hearts (Fig. 4e, f) though the $V_{1/2}$ for channel activation differed ($P < 0.05$) somewhat between groups (Fig. 4g, −46.37 ± 0.59 mV for control cells; −42.03 ± 0.43 mV for Tmem65 KD cells). No other differences in channel gating properties were detected (Table 4). As discussed below (See "Discussion"), the lack of concordance between the $I_{Na}^+$ results with the expression and immunohistochemical measurements is not unexpected, due to disruption and remodeling of ICDs during enzymatic isolation of cardiomyocytes[25,26].

**Tmem65-β1 interaction is required for preserving the perinexus in mouse hearts**
The underlying mechanisms whereby Tmem65 reduction leads to reduced Cx43 as well as NaV1.5 and β1 are unclear. Interestingly, previous interactome studies suggest a putative interaction between Tmem65 and β1[27,28]. Consistent with the studies, super-resolution microscopy in the enface orientation of ICD showed that Tmem65 and β1 were co-localized in normal mouse hearts (Fig. 5a, PCC 0.63 ± 0.05). Moreover, co-immunoprecipitation assays revealed that β1 antibody, but not control sera, successfully co-precipitated with Tmem65 in mouse hearts (Fig. 5b). Reverse co-immunoprecipitations in transfected HEK-293 cells further confirmed these findings (Fig. 5c), which suggest that Tmem65 and β1 physically interacted in mouse cardiomyocytes. Next, we sought to determine if, and to what degree, Tmem65 silencing would impact the perinexal nanodomain, the subcellular ICD domain where β1 and NaV1.5 are located[4]. Transmission electron microscopy showed abnormal structural changes of perinexus in Tmem65 KD hearts in comparison to control hearts (Fig. 5d, highlighted in yellow). Further characterization showed Tmem65 silencing ($P < 0.01$) significantly increased the perinexal intermembrane distance (25.97 ± 11.74 nm in Tmem65 KD hearts; 18.73 ± 7.13 nm in control hearts) but did not alter non-perinexal intermembrane distance within ICDs (Fig. 5e). As discussed further below, these data support the conclusion that Tmem65-β1 interaction is critical for supporting the perinexus in the cardiac ICD.

**Reduced Tmem65 leads to changes in APs and ion fluxes in mouse adult cardiomyocytes**
Our results so far have established that Tmem65 is indispensable for maintaining proper cardiac ICD structure and function as well as cardiac conduction velocity in mouse hearts. Since cardiomyopathy is often associated with changes in action potential (AP) profiles as a result of ion channel remodeling, we compared QT intervals and APs between Tmem65 KD and control hearts. Because heart rates estimated from ECGs (Table 3) were markedly slower ($P < 0.01$) in conscious Tmem65 KD mice (347 ± 16 bpm) compared to conscious control hearts (415 ± 10 bpm) and because QT intervals depend strongly on heart rate, it was necessary to calculate corrected QT intervals (determined with a modified Bazett's formula: $QT_c = QT/sqrt(RR/100)^{1/2})$[29] before comparing QT intervals between the groups. We found that $QT_C$ intervals were longer ($P < 0.01$) in Tmem65 KD hearts (54.30 ± 1.94 ms) than control hearts (45.2 ± 2.0 ms). Consistent with prolonged $QT_C$, APs measured from optical mapping measurements in isolated hearts were prolonged ($P < 0.05$) at 25% repolarization ($APD_{25}$, 17.7 ± 1.8 ms in control hearts; 21.8 ± 1.0 ms in Tmem65 KD hearts) and at 50% repolarization ($APD_{50}$, 26.1 ± 1.4 ms in control hearts; 30.8 ± 1.0 ms in Tmem65 KD hearts) when paced at 11 Hz while no differences were observed without pacing of isolated hearts

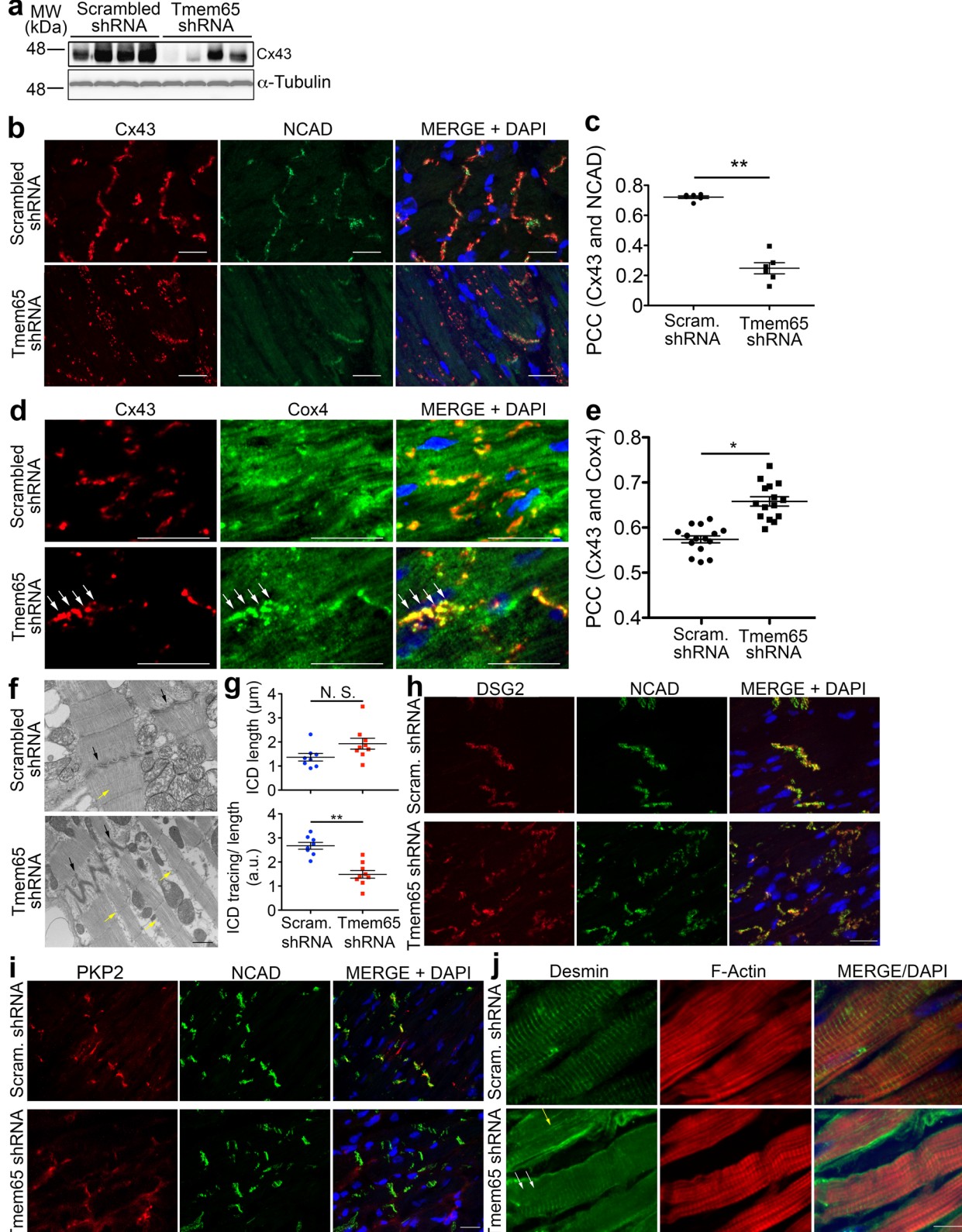

(Table 2). Interestingly, there was a marked delay in the time-to-peak of APs in isolated Tmem65 KD hearts (9.8 ± 0.2 ms in control hearts; 11.8 ± 0.4 ms in Tmem65 KD hearts), possibly related to reduced Cx43 gap junction or NaV1.5 channels, or both components at the ICDs (Figs. 3b and 4b).

To explore the basis for the differences in AP durations, we further examined Ca²⁺ and K⁺ membrane currents in isolated cardiomyocytes.

We found (Fig. 6a–d) that the peak $I_{ca}$ current density was significantly reduced ($P < 0.05$) in Tmem65 KD cells (−5.10 ± 0.35 pA/pF) compared to the control cells (−7.73 ± 0.02 pA/pF) in conjunction with small rightward shifts ($P < 0.01$) in $V_{1/2}$ for channel activation (−13.82 ± 0.32 mV in control cells; −11.33 ± 0.24 mV in Tmem65 KD cells). A significant increase ($P < 0.05$) in the activation slope of $I_{Ca}$ (5.83 ± 0.28 in control cells; 6.86 ± 0.22 in Tmem65 KD cells) and a

**Fig. 3 | Tmem65 KD is associated with ICD defects in mouse hearts.**
**a** Immunoblots demonstrating changes of Cx43 protein levels in Tmem65 KD hearts. α-Tubulin was included as a loading control. **b** Immunofluorescence of Cx43 in control and Tmem65 KD hearts. Internalized Cx43 proteins were seen in Tmem65 KD mouse hearts co-localized with ICD marker NCAD in control hearts (scale bar = 20 μm). **c** Pearson correlation coefficient (PCC)-based quantification showing reduced correlation between Cx43 and NCAD in Tmem65 KD hearts. **P < 0.01. **d** Immunofluorescence of Cx43 and Cox4 in control and Tmem65 KD hearts. Co-localization of Cx43 and Cox4 was visualized (white arrows). Scale bar = 50 μm. **e** Pearson correlation coefficient (PCC)-based quantification showing increased correlation between Cx43 and Cox4 in Tmem65 KD hearts.
**f** Representative transmission electron microscopy of Tmem65 KD (bottom panel) and control hearts (top panel). Aberrant intercalated discs (black arrows) and dismantled myofibers (yellow arrows) were found in Tmem65 KD hearts.

Scale bar = 500 nm. **g** Quantification of ICDs in electron micrographs showing the tendency in increasing ICD length (upper panel) and a significant decrease in ICD curvature smoothness (bottom panel). N.S., Not Significant; **P < 0.01. **h** Immunofluorescence of DSG showing no changes of DSG localization with the ICD marker NCAD in Tmem65 KD hearts. **i** Immunofluorescence of PKP2 showing reduced localization with the ICD marker NCAD in Tmem65 KD hearts. Scale bar = 50 μm. **j** Immunofluorescence of desmin and F-actin in control (upper panels) and Tmem65 KD (lower panels) hearts. Reduced sarcomeric desmin (white arrows) or longitudinal desmin (yellow arrow) were found in Tmem65 KD hearts. Scale bar = 50 μm. Experiments were performed in mice of both genders. Experiments were performed in greater than 3 mice of both sexes. Statistical analyses were performed by one-way ANOVA with Tukey's post-hoc test. Data were expressed as mean ± standard error of the means.

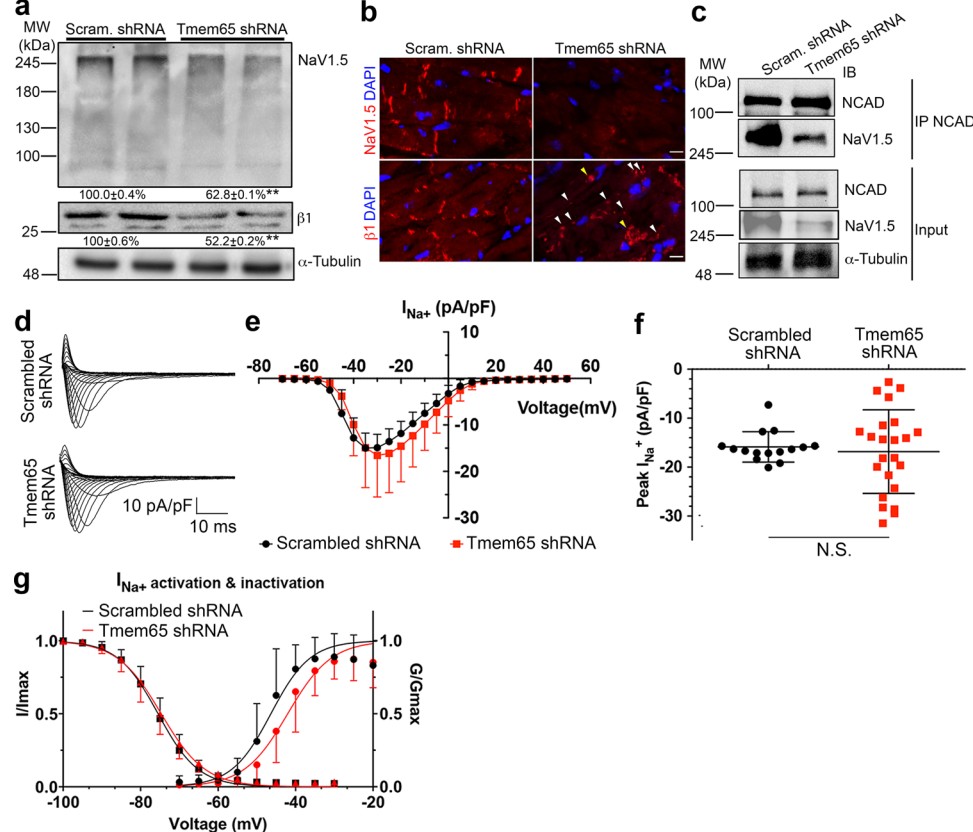

**Fig. 4 | Tmem65 KD leads to losses of NaV1.5 channels at ICDs. a** An immunoblot showing reduced NaV1.5 and β1 protein levels in Tmem65 KD mouse hearts when compared to control samples ($n = 4$ hearts). **P < 0.01. Statistical analysis was performed by one-way ANOVA with Tukey's post-hoc test. Data were expressed as mean ± standard error of means. **b** Immunofluorescence of ICD regions in scrambled and Tmem65 KD hearts, showing the loss of ICD-bound NaV1.5 in Tmem65 KD hearts. β1 staining showed scattered puncta (white arrows) or aggregates (yellow arrows) also in Tmem65 KD hearts. **c** Immunoprecipitation assays using NCAD antibody and mouse cardiac lysates showed a reduced NaV1.5/NCAD interaction in Tmem65 KD mouse hearts. **d** Representative images of sodium current ($I_{Na}^{+}$) in control vs. Tmem65 KD cardiomyocytes. **e** Current density-voltage (I-V) plot shows

that $I_{Na}^{+}$ density did not significantly differ between scrambled (black) and Tmem65 KD (red) cardiomyocytes. **f** Quantification of peak $I_{Na}^{+}$ density in scrambled and Tmem65 KD cardiomyocytes. No statistical difference was found. **g** Voltage-dependence of $I_{Na}^{+}$ activation and steady-state inactivation in cardiomyocytes with the least-square fits to the Boltzmann function. A statistical significance was found between Tmem65 KD and control cardiomyocytes in activation state, but not in inactivation state. All measurements are summarized in Table 4. $N = 16$ control cells and 25 Tmem65 KD cells for in $I_{Na}^{+}$. All cells were isolated from male mouse hearts. For patch-clamp experiments, data were presented as mean ± standard deviation. The difference of current density was compared using unpaired student t-test.

reduction in the channel inactivation of $I_{Ca}$ (−27.75 ± 0.39 mV in control cells; −23.88 ± 0.18 mV in Tmem65 KD cells) were also found in Tmem65 KD hearts (Fig. 6d, Table 4). These results collectively demonstrate that loss of Tmem65 affects $Ca^{2+}$ fluxes in mouse adult cardiomyocytes.

K⁺ currents were also affected by the loss of Tmem65 in adult cardiomyocytes (Fig. 6e). In particular, the peak outward K⁺ current densities were increased ($P < 0.05$) in Tmem65 KD (41.5 ± 2.45 pA/pF)

cardiomyocytes compared with control cardiomyocytes (30.44 ± 1.86 pA/pF) (Fig. 6f, and Table 4). While differences in peak K⁺ and $Ca^{2+}$ currents between the groups seem inconsistent with the slight AP prolongation seen in Tmem65 KD myocardium, it is important to recognize that mouse cardiomyocytes have at least 5 different voltage-gated K⁺ currents with unique inactivation kinetic properties[30] that can have distinct effects on AP profile. Accordingly, we using kinetic fitting procedures[31] that the fast delayed rectifier K⁺ currents ($I_{K,slow1}$)

**Table 4 | Whole-cell voltage clamping of control and Tmem65 KD cardiomyocytes**

| Unite (pA/pF) | Scrambled shRNA ($n = 16$) | Tmem65 shRNA ($n = 25$) |
|---|---|---|
| Peak $I_{Na}^+$ | $-15.87 \pm 0.80$ | $-16.83 \pm 1.78$ |
| $I_{Na}^+$ activation (mV) | $-46.37 \pm 0.59$ | $-42.03 \pm 0.43$* |
| $I_{Na}^+$ activation slope | $5.06 \pm 0.52$ | $5.63 \pm 0.38$ |
| $I_{Na}^+$ inactivation (mV) | $-75.51 \pm 0.21$ | $-74.70 \pm 0.19$ |
| $I_{Na}^+$ inactivation slope | $-5.20 \pm 0.19$ | $-5.67 \pm 0.16$ |
| Peak $I_{Ca}^{2+}$ (pA/pF) | $-7.73 \pm 0.22$ | $-5.10 \pm 0.35$ |
| $I_{Ca}^{2+}$ activation (mV) | $-13.32 \pm 0.32$ | $-11.33 \pm 0.24$* |
| $I_{Ca}^{2+}$ activation slope | $5.83 \pm 0.28$ | $6.86 \pm 0.22$* |
| $I_{Ca}^{2+}$ inactivation (mV) | $-27.75 \pm 0.39$ | $-23.88 \pm 0.18$* |
| $I_{Ca}^{2+}$ inactivation slope | $-6.67 \pm 0.25$ | $-6.54 \pm 0.16$ |

| Unit (pA/pF) | Scrambled shRNA ($n = 10$) | Tmem65 shRNA ($n = 10$) |
|---|---|---|
| Total $I_{K}^+$, outward | $30.44 \pm 1.86$ | $41.50 \pm 2.45$* |
| Total $I_{to}$ | $9.81 \pm 1.85$ | $25.25 \pm 2.39$* |
| Total $I_{K}^+$, slow1 | $8.19 \pm 1.19$ | $5.13 \pm 0.77$* |
| Total $I_{K}^+$, slow2 | $8.15 \pm 0.57$ | $7.51 \pm 0.73$ |
| Total $I_{ss}$ | $4.02 \pm 0.35$ | $4.25 \pm 0.46$ |
| $I_{K+, outward}$ (4-AP-sensitive) | $16.6 \pm 0.96$ | $18.45 \pm 1.12$ |
| $I_{to}$ (4-AP-sensitive) | $7.21 \pm 1.28$ | $15.96 \pm 1.19$* |
| $I_{K}^+$, slow1 (4-AP-sensitive) | $6.24 \pm 0.96$ | $1.36 \pm 0.47$** |
| $I_{K}^+$, slow2 (4-AP-sensitive) | $4.40 \pm 0.46$ | $3.18 \pm 0.50$ |
| $I_{ss}$ (4-AP-sensitive) | $0.07 \pm 0.05$ | $0.375 \pm 0.11$* |
| $I_{K}^+$, outward (4-AP-insensitive) | $15.00 \pm 2.01$ | $22.60 \pm 1.90$* |
| $I_{to}$ (4-AP-insensitive) | $3.55 \pm 0.84$ | $8.95 \pm 1.31$* |
| $I_{K}^+$, slow1 (4-AP-insensitive) | $3.35 \pm 0.57$ | $4.59 \pm 0.56$** |
| $I_{K}^+$, slow2 (4-AP-insensitive) | $4.40 \pm 0.60$ | $5.62 \pm 0.52$ |
| $I_{ss}$ (4-AP-insensitive) | $4.42 \pm 0.83$ | $3.84 \pm 0.37$ |

$n = 16$ control and 25 Tmem65 KD cardiomyocytes for in $I_{Na}^+$ and $I_{Ca}$ measurements. $n = 10$ cardiomyocytes per group for $I_k^+$ measurement. All cells were isolated from male mouse hearts.
*mV* millivolt, *4-AP* 4-aminopyridine.
*$p < 0.05$, **$p < 0.01$.

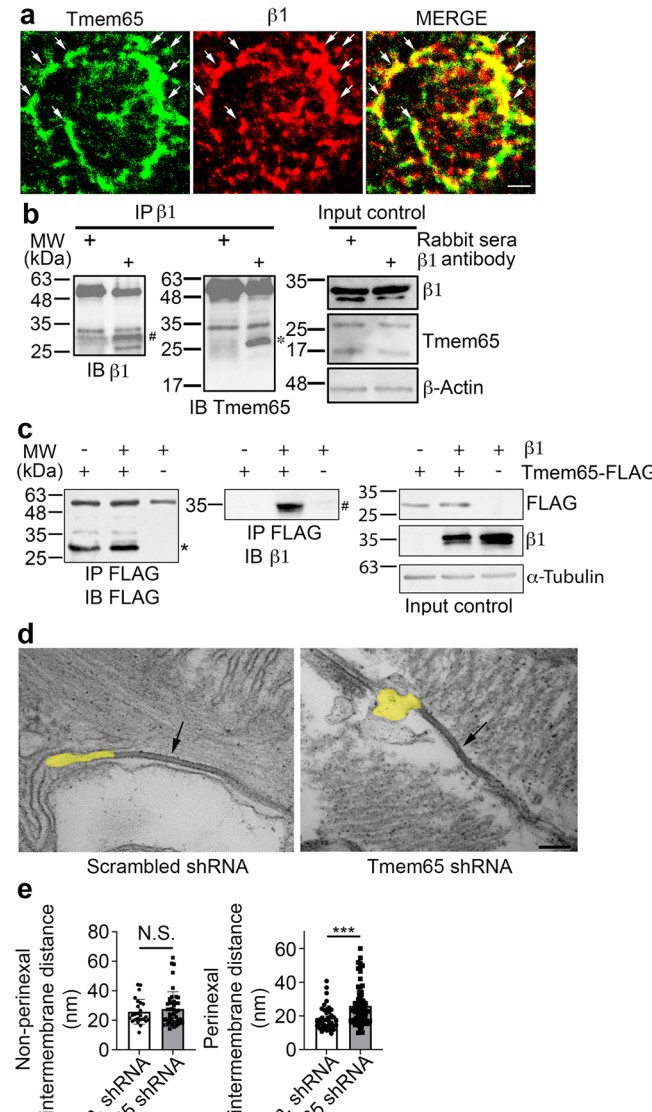

**Fig. 5 | Tmem65-β1 interaction is required for preserving the perinexus in mouse hearts. a** Representative super-resolution microscopy showing Tmem65 (left panel, green) and β1 (mid panel, red) co-localized (right panel, yellow) in the enface orientation of the ICD in mouse hearts. Scale bar = 1 μm. Five images were taken from 2 mouse hearts for determining PCC. **b** Co-immunoprecipitation assays using β1, showing that Tmem65 was co-precipitated (*) from mouse hearts by β1 antibody (#). **c** Co-immunoprecipitation assays showing that Tmem65-FLAG (*) co-precipitated β1 (#) in transfected HEK293T cells. **d** Representative transmission electron microscopy showing an aberrant perinexal domain in the Tmem65 KD heart. Perinexus next to the Gap junction (black arrow) was highlighted in yellow. Scale bar = 100 nm. **e** Quantification of intermembrane distances at perinexal junctions and non-perinexal regions within ICDs in Tmem65 KD and control hearts. Tmem65 silencing led to increased perinexal intermembrane distance (right panel) in comparison to control hearts, but did not affect non-perinexal intermembrane distances (left panel). ***$P < 0.01$. Experiments were performed in mice of both sexes. Experiments completed in 4 Tmem65 KD hearts and 3 control hearts of both sexes. For the image analysis of the EM images, nested *t*-test comparisons between control and treatment were carried out using the log-transformed data set. Data were presented as mean ± standard deviation.

(Fig. 6g), encoded by Kv1.5, was reduced ($P < 0.05$) in Tmem65 KD cardiomyocytes ($5.13 \pm 0.77$ pA/pF) compared to control ($8.19 \pm 1.19$ pA/pF) while neither slow delayed rectifier ($I_{k,slow2}$) (Fig. 6h) encoded by Kv2 channels, nor the sustained current ($I_{ss}$) (Fig. 6i) differed ($P > 0.5017$) between the groups. On the other hand, the total transient outward current, $I_{to}$ (Fig. 6k) was increased ($P < 0.01$) in Tmem65 KD cardiomyocytes ($25.25 \pm 2.39$ pA/pF) when compared to control cardiomyocytes ($9.81 \pm 1.85$ pA/pF in control cells) which explains the increased peak outward K$^+$ current. But it is important to appreciate that $I_{to}$ is comprised of two components (i.e. fast $I_{to}$ called $I_{to,f}$ and slow $I_{to}$ called $I_{to,s}$) with very different rates of recovery from inactivation. Consequently, only fast recovering $I_{to,f}$ impacts markedly on AP duration. By exploiting the differential sensitivities of $I_{to,f}$ and $I_{to,s}$ to 4-AP (Fig. 6j), we found that both $I_{to,f}$ (Fig. 6l) and $I_{to,s}$ (Fig. 6m) were increased in the Tmem65 KD cardiomyocytes (Table 4). Parenthetically, evaluation of the 4-AP-sensitive currents also allowed a more accurate quantification of $I_{K,slow1}$ which was found to be reduced by ~4.5-fold by Tmem65 KD (Fig. 6n, Table 4). These ion current measurements support the conclusion that the small AP prolongations seen in Tmem65 KD cardiomyocytes arise from an overwhelming reduction in repolarizing currents provided by $I_{K,slow1}$ in the face of increases in $I_{to}$-based currents and reductions in $I_{Ca}$ (See "Discussion").

## Discussion

In this study, we have established a mouse model for investigating Tmem65 function in mouse hearts via rAAV9 virus-mediated gene silencing. Both immunoblots and qPCR demonstrated that a single intraperitoneal injection of AAV9 results in a > 85% reduction of Tmem65 protein and mRNA levels in mouse hearts 3 weeks post viral injection. This reduction in Tmem65 leads to clear signs of congestive heart failure (CHF)−including dilated ventricle with fibrosis, reduced cardiac output, increased expression in protein stress markers - by

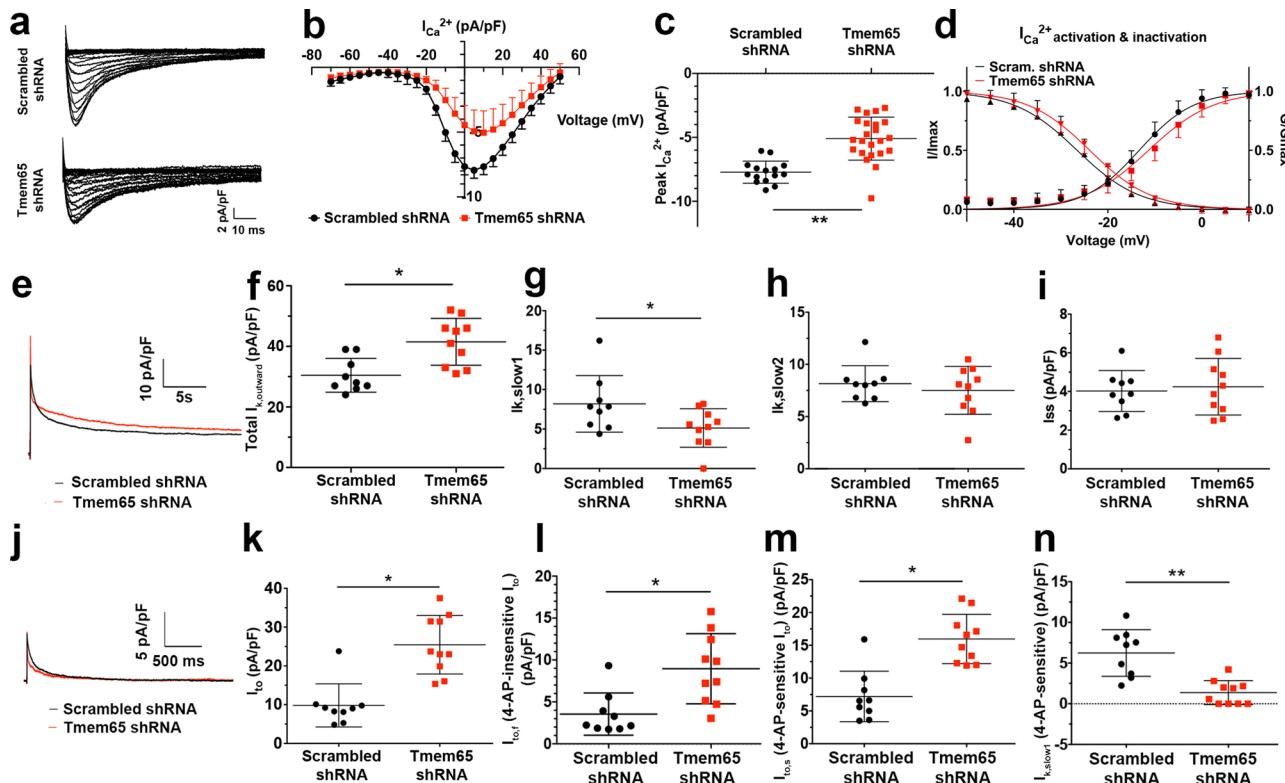

**Fig. 6 | Tmem65 KD leads to changes in APs and ion fluxes in mouse adult cardiomyocytes.** All measurements were recorded by whole-cell voltage clamping in cardiomyocytes, isolated from scrambled (black) and Tmem65 KD cells (red). **a** Representative images of calcium current ($I_{Ca}$) in cardiomyocytes. **b** Current density-voltage (I–V) plot shows that $I_{Ca}$ density significantly differ between −20-40 mV voltage in scrambled and Tmem65 KD cardiomyocytes. **c** Quantification of peak $I_{Ca}$ density in scrambled and Tmem65 shRNA cardiomyocytes. $I_{Ca}$ was significantly reduced in Tmem65 KD cardiomyocytes in comparison to control cells ($P < 0.01$). **d** Voltage-dependence of $I_{Ca}$ activation and steady-state inactivation in cardiomyocytes following the least-square fitting to the Boltzmann function. Statistical significance was only found in $I_{Ca}$ activation and slop between Tmem65 KD and control cardiomyocytes. **e** Representative recordings of outward $I_k^+$ in scrambled (black) and Tmem65 KD (red) mice. **f** Quantification for total outward currents ($I_{k, outward}$) was significantly higher in Tmem65 KD cells.

$*P < 0.05$. **g** Quantification for, $I_{k, slow1}$ was reduced in Tmem65 KD cardiomyocytes. $*P < 0.05$. **h** Quantification for $I_{k, slow2}$. **i** Quantification for remaining current, $I_{ss}$. **j** Representative recordings of 4-AP sensitive outward $I_k^+$ following 4-AP administration in scrambled (black) and Tmem65 KD (red) mice. **k** Quantification for transient outward currents ($I_{to}$) was higher in Tmem65 KD cells. **l** Quantification of 4-AP-insenitive $I_{to, fast}$ ($I_{to,f}$) showing increased current in Tmem65 KD cells. $*P < 0.05$. **m** Quantification of 4-AP-sensitive $I_{to, slow}$ ($I_{to,s}$) showing increased current in Tmem65 KD cells. $**P < 0.01$. **n** Quantification for 4-AP sensitive $I_{k, slow1}$ was significantly reduced in Tmem65 KD cells. All measurements are summarized in Table 4. $n = 16$ control cells and 25 Tmem65 KD cells for $I_{Ca}$ measurements. $n = 10$ cells per group for $I_k^+$ measurement. All cells were isolated from male mouse hearts. Data were presented as mean ± standard deviation. The difference of current density was compared using unpaired student $t$-test.

6−7 weeks post injection, with all mice expiring 7 weeks post injection. To explore the pathogenesis of the CHF, we characterized mice after 3 weeks of Tmem65 silencing because at this point the Tmem65 KD mice appeared behaviorally indistinguishable from control and mice started dying thereafter. After 3 weeks hearts with Tmem65 silencing showed mild eccentric hypertrophy characterized by slight ventricular lumen dilation without increases in wall thickness. These changes were accompanied by increases in both the length and width of cardiomyocytes which was associated with destabilization of the structural triad at the ICD. Such ICD changes are expected to impede mechanical coupling between cardiomyocytes, which could underlie the reductions in sarcomeric desmin and dismantled myofibrils seen in Tmem65 KD hearts. Similar ICD changes with desmin have been reported previously in patients and transgenic mice and been attributed to myofibril breakdown[22,32]. These observations help to explain the cardiomyopathy and impaired contractility seen in the Tmem65 mice. Moreover, desmosome junctions can inhibit RAS/MAPK/ERK signaling pathway in the heart[33] which would help explain the increased ERK1/2 phosphorylation and thereby cardiac hypertrophy seen with Tmem65 KD[34,35]. The combination of sarcomeric breakdown, and ERK1/2 activation in association with ICD structural changes may contribute to progression of Tmem65 KD towards a cardiomyopathy phenotype,

which is characterized classically by sarcomere disarray and ICD defects[33].

Another prominent functional consequence of the Tmem65 KD hearts was slowed ventricular conduction, associated with marked prolongation of the QRS duration as well as PR interval delays, indicative of impaired atrioventricular conduction. Slowed cardiac conduction is consistent with marked structural remodeling of ICDs with Tmem65 KD, characterized by zig-zag ICD pattern and the mislocalization of proteins at junctional regions. Presumably these changes were related to the reduced and internalized Cx43 protein as well as the loss of Nav1.5, which offer explanations for slowed conduction[36-38]. These defects were also associated with a widening of perinexus intermembrane distance with changes in non-perinexal distances, a phenomenon that has been linked to arrhythmogenic conduction defects[4]. Impaired conduction could further originate from changes in ICD and perinexal nanodomain structure[39-42] which can modulate interplay between Cx43-dependent gap junctions and trans-sarcolemmal ion fluxes, and further strongly influence the localized transmembrane voltages and thereby NaV1.5 function[36,38]. Indeed, since we observed no $I_{Na}$ changes between Tmem65 KD hearts and control cardiomyocytes, it would seem reasonable to suggest that any effects of $I_{Na}$ on slowed conduction in Tmem65 KD hearts would be

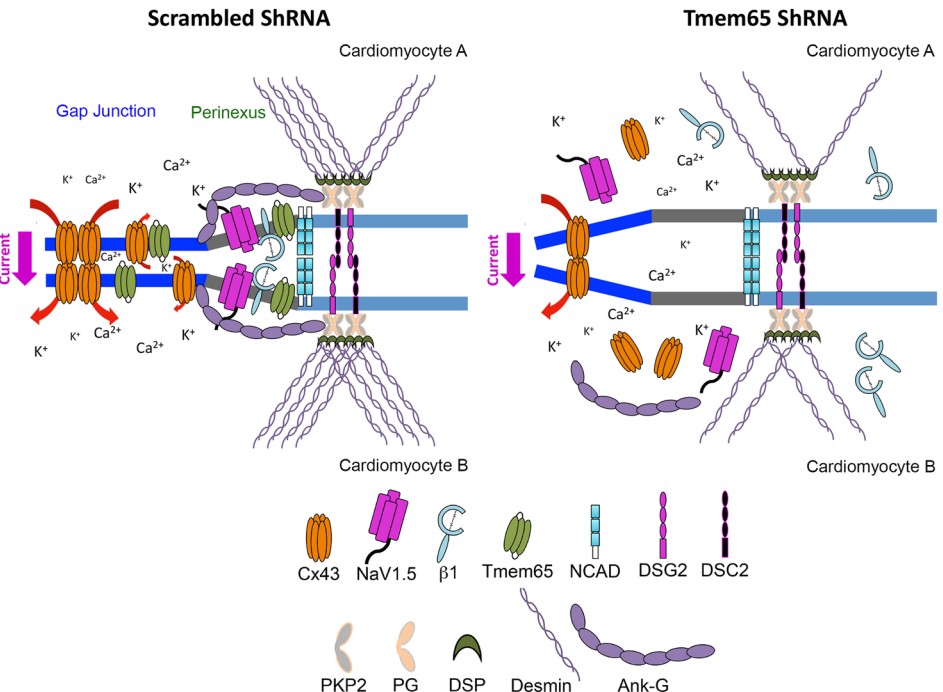

**Fig. 7 | Schematic illustration for the role of Tmem65 in mouse hearts and the summary of this study.** In control cardiomyocytes, Tmem65-β1 interaction is responsible for stabilizing the perinexus in the ICD and is needed for Cx43 and NaV1.5 localization to the ICD (image on the left). Conversely, reduced Tmem65 is associated with destabilized perinexus, reduced Cx43 and NaV1.5 at the ICD (image on the right). Desmosomes were also impaired in Tmem65 KD hearts. Specifically, the striated pattern of intermediate filament desmin and the localizations of desmosome proteins DSP and DSG2 to ICDs are markedly reduced in Tmem65 KD hearts.

secondary to changes in the ICD. However, it is worth mentioning that in our previous results wherein ICD structure was disrupted with β1 interference peptides, we also observed no change in whole-cell $I_{Na}$, despite a similar loss of Nav1.5 at the perinexus and selectively reduced perinexal $I_{Na}$[4]. These observations can be readily explained by previous electron micrographic analyses of isolated cardiomyocytes showing that enzymatic digestion leads to abrupt "back-folding" of membranes covering the ICD which offers protection from the effects of gap junctional hemichannels[25,26], while also expected to eliminate the function of perinexal NaV1.5 channels during whole-cell recordings. This disparity between immunohistochemical and patch-clamp results might also be important generally to consider when assessing other cardiac sarcolemma channels that are localized at the intercalated disks such as $I_{to,f}$ in atria[43] and $I_{K1}$ (i.e. inward rectifier $K^+$ channels)[44]. Future studies with advanced gap junction-specific patch-clamp technology will help to investigate the functional impact of the ICD-bound NaV1.5 as well as other channels in response to Tmem65 silencing. Together, our findings indicate a novel function of Tmem65 that is critical for fine-tuning cardiac electrophysiology in a Cx43-dependent manner.

While the basis for the disruption of ICD and perinexus structure and function is unclear, we did identify a novel interaction between Tmem65 and β1 which could interrupt β1-mediated cell adhesion at the perinexus nanodomain. These data are not only consistent with a previous finding of perinexus-localized Cx43 (or NaV1.5) function at the ICD[14], but they also uncover a new role for Tmem65 in the heart. Since NaV1.5/β1- (ephaptic) and Cx43-enriched (gap) junctions overlap at the perinexal nanodomain[4], a further understanding of protein complexes in this region and their function will help to elucidate perinexal biology. In this regard, our studies suggest Tmem65-β1 interaction is critical for maintaining the perinexal structure in cardiomyocytes. Moreover, Tmem65 also interacts, and co-localizes, with Cx43 in mouse myocardium[16], but its interaction with NaV1.5 could never be detected by co-immunoprecipitation assays. When this observation is combined with the inability of β1 to directly interact

with NaV1.5 channels as demonstrated by x-ray crystallography[15], we are led to speculate that NaV1.5 and Tmem65 interact via intermediate proteins at the ICD. Cx43 appears to be a reasonable candidate as it has been shown to independently interact with NaV1.5[10] and Tmem65[16] in mouse hearts, although the molecular mechanism remains mostly unknown. Such a scenario would align with the ability of Cx43 mutants to prevent NaV1.5 localization to the ICD[45]. Moreover, Marchal et al. reported that end binding protein 1 (EB1) is required for NaV1.5 adhesion to the ICD[46] and Cx43 localization to the ICD also depends on EB1[12]. Another potential factor contributing to the changes in ICD structure seen following Tmem65 KD is Ank-G, which is an indispensable scaffold protein that strengthens Cx43 and plakophilin 2 localization to the ICD[10]. The recruitment of Ank-G to the perinexus requires *trans*-homophilic adhesion to β1[47] and Ank-G ablation in mice leads to disruption of desmosomes and gap junctions[11,48], as we observed in Tmem65 KD hearts. Thus, our findings support the model depicted in Fig. 7 wherein desmosome-gap junction-ephaptic junction form a functional triad for providing mechanical and electrochemical couplings[10,11,48] whose integrity depends critically on Tmem65-β1 interactions. Future studies will be required to assess trafficking and the interactome underpinning of Tmem65-β1 complex to the ICD as well as the impact on ICD and perinexus structure and function. Such studies could also further our understanding of the impact of these changes on cardiac electrophysiology and arrhythmic conditions.

Although APDs measured using optical mapping recordings did not differ between Tmem65 KD hearts and control in sinus rhythm after 3 weeks of Tmem65 KD, APDs were prolonged in Tmem65 KD mice when the heart was paced at 11 Hz consistent with prolonged corrected QT (QT$_C$) intervals, as is commonly seen in cardiomyopathies and heart disease[49]. Somewhat unexpectedly, the APD prolongation observed in Tmem65 KD myocardium was associated with reductions in depolarizing $I_{Ca}$ and increases in the peak $K^+$ currents. However, further quantification of the various voltage-dependent $K^+$ currents using kinetics analyses of $K^+$ current decay and 4-AP sensitivities[31] revealed that Tmem65 KD caused a more than 3-fold

reduction in $I_{K,slow1}$, which is a major repolarizing current that is active throughout the AP duration in mouse myocardium[50]. On the other hand, cardiomyocytes from the Tmem65 KD hearts displayed increases in $I_{to,f}$ which contributes primarily to early repolarization in the mouse heart[51,52]. The slow component of $I_{to}$ (i.e. $I_{to,s}$) was also increased following 3 weeks of Tmem65 silencing, but this is unlikely to influence APD because the slow recovery from inactivation of $I_{to,s}$ limits current magnitude in the rapidly beating mouse heart[51,52]. Regardless, the APD prolongation, while very modest, is consistent generally with previous studies showing APD prolongation in cardiomyopathy and heart disease. However, APD prolongation is often associated with reductions in $I_{to,f}$[52]. The absence of $I_{to,f}$ reduction in the Tmem65 hearts may reflect the short-term nature of our studies as well as the use of young mice, which may not recapitulate the known changes in ion channel expression and activity with chronic disease[53]. Indeed, consistent with our findings of modest electrical remodeling at 3 weeks, a previous study showed that in a mouse model of DCM, minimal changes in APD were observed at 1 month of disease, with changes in ion currents, APD prolongation, and increased mortality being more pronounced at 3 months of disease[54].

Interestingly, consistent with reductions in Cx43, reduced $Ca^{2+}$ current and altered $K^+$ currents in Tmem65 KD myocytes is consistent with reports that antagonizing Cx43 junction leads to changes in $Ca^{2+}$ and $K^+$ fluxes at the perinexal nanodomain in cardiomyocytes[36,38]. In addition, $Ca^{2+}$ transients in Tmem65 KD cardiomyocytes became irregular and non-quantifiable when cells were paced at 10 Hz (Supplementary Fig. 6), which, along with reduced $Ca^{2+}$ currents, suggests a reduction in $Ca^{2+}$-induced $Ca^{2+}$ release. Moreover, while the relevance of reduced $Ca^{2+}$ currents to ventricular conduction slowing would be minimal, it could contribute to prolonged PR intervals as $I_{ca}$ is crucial for (slow) conduction through the AV node[55]. Nonetheless, whether Tmem65 KD directly impacts on $Ca^{2+}$ and $K^+$ currents versus indirect effects arising from the induced heart disease[7,56] is unclear.

Our study is not without limitations. One limitation to this study is the inability to capture arrhythmia on ECG in Tmem65 KD mice and is primarily due to several technical challenges. The first challenge is that the standard ambulatory ECG devices designed for adult mice are too big for the Tmem65 KD mice that experience sudden cardiac death 3 weeks post injection. Another challenge is that the use of anesthesia reduces cardiac output by slowing down the heart rates of knockdown mice that already have slower heart beat. Indeed, many Tmem65 KD mice died in the anesthesia chamber. Although the presence of arrhythmia remains to be determined, our study has demonstrated, with multiple lines of evidence, that Tmem65 KD is associated with the arrhythmogenic nature in the heart. These evidences include, but not limited to, reduced Cx43 (and NaV1.5) protein levels, structural changes at the ICD (electron micrographs and immunofluorescence, showing changes in gap, ephaptic, and desmosome junctions), and reduced conduction velocity in mouse hearts. One alternative for overcoming this limitation and for confirming a causative relationship to arrhythmia would be to establish ambulatory ECG in adult mice following disrupting Tmem65 expression.

As noted, we observed a differential impact of Tmem65 function between sexes, with most male mice died in 3 weeks following gene silencing, while female mice survived until 6–7 weeks. While the underlying mechanism remains unknown, Foulds et al. reported that the human steroid receptor RNA activator modulated Tmem65 expression in MCF-7 cells in response to estradiol[57]. From that finding, one might hypothesize that the estradiol cycle and activation of endogenous Tmem65 expression may delay pathological progression, though it would be difficult to visualize this sex-based KD in knockout female mice, consistent with decreased cardiovascular mortality seen in women with DCM[58]. Nonetheless, our observations are similar to another mouse model of ICD defects[59] in which males disproportionately develop DCM compared to females.

It seems reasonable to suggest that the mortality seen following Tmem65 silencing represents sudden cardiac death due to arrhythmias[60] either because of the arrhythmogenic substrate arising from slow cardiac conduction as a result of aberrant ICD (reduced gap junctional channels, impaired perinexus function, reduced junctional Nav1.5 channels) or fibrosis related to DCM. Alternatively, given the profound reductions in echocardiographically derived functional indices mice might also have died due to progressive heart failure. Unfortunately, we were unable to determine the cause of death in our mice due to their young age (and therefore small size) which precluded the use of telemetry recordings. We were also unable to make echocardiographic assessment measurements at 3-weeks which precluded us from directly evaluating the progression of heart disease. Thus, future studies will be needed to understand the sex-dependent cause of death that occurs with reduced Tmem65 levels.

## Methods

### Plasmid constructs and chemical reagents

Our previously published Tmem65 shRNA (5'-CCAGGACAGCTGAGA-TATGTA-3') in a miR30-like format was transferred from pLKO1 plasmids to AAV plasmid containing a cardiac troponin T (cTnT) promoter (pAcTnT; gift from Dr. Brent French, University of Virginia)[61]. Proteasome inhibitor MG-132 (Cell Signaling Technology, 2194 S) stock was prepared in DMSO to the final concentration of 10 mM. The lysosome inhibitor, chloroquine, (Sigma-Aldrich, C6628) was prepared in ddH$_2$O to the final concentration of 50 mM. Transfection reagent polyethylenimine (PEI, 23966-1) was acquired from Polysciences, Inc (Warrington, PA) and prepared in ddH$_2$O, adjusted to pH 7.0 with HCl, and final concentration of 1 mg/ml. The final solution was filtered through 0.22 μm membrane, aliquoted, and stored at −80 °C.

### rAAV packaging, purification, and tittering

rAAV9 viral particles harboring Tmem65 shRNA (or scrambled shRNA) were prepared in HEK cells[61,62]. Briefly, HEK293T cells were transfected with AAV-Tmem65 shRNA (or AAV-scrambled shRNA) and pDG9 plasmids (gift from Dr. Roger Hajjar, modified from[63]). Three days post transfection, cells were collected in lysis buffer (50 mM Tris-HCl; pH8.5, 50 mM NaHCO$_3$) and were frozen and thawed three times. Isolated cell crude extract was treated with benzonase (Sigma, E1014) at 37 °C for 30 min. Virions were purified by discontinuous iodixanol gradient-based ultracentrifugation and viral titers determined by qPCR[61,62].

### Animals

All experiments involving animals were conducted in accordance with the Institutional Animal Care and Use Committee of the University of Toronto (University of Toronto Local Animal Care Committee) and York University (The Animal Care Committee). CD1 mice were purchased from Charles River Laboratory (Wilmington, MA). $4.0 \times 10^{11}$ rAAV9 viral genomes were delivered intramyocardially or intraperitoneally per mouse pup of 3–7 days post birth. Injected animals were observed for any signs of distress for 48 h before returning to a regular animal facility.

### Immunoblots

Freshly isolated mouse ventricles were pulverized in liquid nitrogen by mortar and pestle. Protein lysates were harvested in radio-immunoprecipitation (RIPA) buffer (20 mM Tris-HCl; pH 7.5, 150 mM NaCl, 1 mM EDTA, 1 mM EGTA, 1% NP-40, 1% sodium deoxycholate, 0.1% sodium dodecyl sulfate) supplemented with protease (Roche, 11836170001) and phosphatase (Pierce, 184724) inhibitors. Protein quantification was carried out by Bradford assays (Sigma, B6916). Standard protein electrophoresis and transfer were performed[16]. Antibodies and their dilution are listed below. Tmem65 antibody (1:1000 dilution; Sigma, HPA025020), Connexin43 antibody (1:1000; Sigma, C6219), α-Sarcomeric Actinin (1:1,000; Sigma, A7811),

**Table 5 | List of qPCR primers. Primers and their sequences are listed below**

|  | Primer sequences |
|---|---|
| Tmem65 | 5'-GGG CAC ACA CCC CAA GAA G-3' |
|  | 5'-ACC CTA CGA AAG GTA TCG CAT G-3' |
| Cx43 | 5'-ACA GCG GTT GAG TCA GCT TG-3' |
|  | 5'-GAG AGA TGG GGA AGG ACT TGT-3' |
| Nppa | 5'-ATC ACC CTG GGC TTC TTC CT-3' |
|  | 5'-TGT TGG ACA CCG CAC TGT AC-3' |
| Nppb | 5'-GAG GTC ACT CCT ATC CTC TGG'–3' |
|  | 5'-GCC ATT TCC TCC GAC TTT TCT C'–3' |
| Hprt1 | 5'-CAA GCT TGC TGC TGA AAA GGA-3' |
|  | 5'-TGA AGT ACT CAT TAT AGT CAA GGG CAT ATC-3' |

These primers were used for quantifying transcript levels of Tmem65, Cx43, Nppa, and Nppb in Tmem65 KD mouse hearts. Hprt1 was included as a loading control.

α-Sarcomeric Actin (1:1,000; Sigma, A2172), MyHC (1:40; DSHB, MF20), β-Actin (1:200; SC-47778, Santa Cruz Biotecholgy), cTnT (1:40; DSHB, CT3), MYH7 (1:40, DSHB, A4.840), Tubulin (1:40; DSHB, E7), FHL1 (1:1000; ab49241), Erk1/2 (1:1,000; Cell Signaling Technology, 9102), Phospho-Erk1/2 (1:1,000; Cell Signaling Technology, 9101), β1 (1:1,000, Cell Signaling Technology), NaV1.5 (1:1000; ASC-005, Alomone Labs), N-Cadherin (1:30 unconcentrated hybridoma supernatant; MNCD2, DSHB), and FLAG (1:1,000; Sigma, F1804).

### Co-immunoprecipitation (Co-IP) assays of β1 and Tmem65 in mouse hearts and transfected HEK-293 cells

Mice were sacrificed by $CO_2$ euthanasia and cervical dislocation. Mouse hearts were perfused with 10 ml ice-cold 1× PBS and snap frozen in liquid nitrogen. Cardiac protein lysates were prepared as following. 100 mg of tissues were homogenized by mortar and pestle in 1 ml immunoprecipitation buffer (20 mM HEPES; pH 7.4, 150 mM NaCl, 5 mM EDTA, 0.5% Triton X-100, supplemented with 2× protease and phosphatase inhibitors). Protein lysates were solubilized on ice for 15 min and insoluble fraction was eliminated by centrifugation at 14,000 × $g$ for 15 min. Protein lysates were pre-cleared with 50 μl of protein A/G agarose (Pierce) for 30 min at 4 °C on an end-to-end rotor. For immunoprecipitation, 1 mg pre-cleared lysate was mixed with 5 μl β1 antibody (Cell Signaling Technology, D4Z2N) or NCAD antibody (Cell Signaling Technology, D4R1H) overnight at 4 °C on an end-to-end rotor. Next day, 40 μl BSA-blocked protein A/G agarose were mixed with the overnight lysate for an hour at 4 °C on an end-to-end rotor. Precipitated proteins were washed three times with ice-cold lysis buffer and once with 1× PBS. Proteins were eluted in 2× Laemmli buffer containing 100 mM DTT and immunoblottings were performed as described above with following antibodies and dilution ratios. Tmem65 antibody (Sigma, HPA025020, 1:1000 dilution (v/v) in 5% non-fat milk in TBST). β1 antibody (Homemade antibody[4], 1:1000 dilution (v/v) in 5% non-fat milk TBST). NaV1.5 antibody (1:1000 dilution (v/v) in 5% non-fat milk TBST).

Co-IP assays were also performed in transfected HEK-293 cells for testing the interaction between FLAG (DYKDDDDK)-tagged Tmem65 and HA (YPYDVPDYA)-tagged β1. Briefly, HEK-293 cells were seeded at 50% confluency in 100-mm dishes a day before transfection. Next day, transfection was performed by mixing 5 μg of each plasmid and 15 μg PEI/plasmid in serum-free media, incubated for 15 min at the room temperature, and added to cells dropwise. Cells were returned to a cell incubator. Next day, cells were given fresh media and returned to a cell incubator for 24 h. On the day of Co-IP assays, cells were washed twice with ice-cold 1× PBS, lysed with immunoprecipitation buffer (same buffer as above), and incubated on ice for 15 min. Soluble proteins were separated by centrifugation at 14,000 × $g$ for 10 min at 4 °C and transferred to a new tube labeling cell-free lysate. FLAG-Tmem65 was immunoprecipitated by mixing 500 μg soluble proteins with 1 μg FLAG antibody (Sigma, F1804) over night at 4 °C on an end-to-end rotor. Next day, 40 μl BSA-blocked protein A/G agarose were mixed with the overnight lysate for an hour at 4 °C on an end-to-end rotor. Precipitated proteins were washed three times with ice-cold lysis buffer and once with 1× PBS. Proteins were eluted in 2× Laemmli buffer containing 100 mM DTT and immunoblottings were performed as described above with following antibodies and dilution ratios. FLAG antibody (1:1000 dilution (v/v) in 5% non-fat milk in TBST); HA antibody (1:1000 dilution (v/v) in 5% bovin serum albumin in TBST).

### qPCR analyses

Three weeks following viral injection, perfused mouse hearts were harvested and stored in liquid nitrogen. Total RNA extraction was performed with TRIzol (Invitrogen) and RNA integrity was confirmed[64]. Reverse transcription-mediated cDNA synthesis (ThermoFisher Scientific, 4387406) and real-time qPCR (RT-qPCR) quantification (ThermoFisher Scientific, 4367659) were performed per manufacturers' instructions and were compliant with the MIQE guidelines[18]. Validated primers are listed in Table 5.

### Histology

Euthanized mice were perfused with 10 ml 1× ice-cold PBS and the hearts were frozen in optimal cutting temperature compound (Fisher Scientific, Ontario, Canada) in dry ice-chilled 2-methylbutane (Sigma, M32631). Cardiac sections were prepared at 5-μm thickness. Standard Mayer's hematoxylin and eosin (H&E) and Masson's Trichrome staining were performed by the Toronto Oral Pathology Service (Faculty of Dentistry, University of Toronto, Toronto). Immunofluorescence was performed[16] using antibodies and their dilution are listed below. Cx43 (1:100; C6219, Sigma), Cox4 (1:50, ab16056, Abcam), NCAD (1:20 unconcentrated hybridoma supernatant; MNCD2, DSHB), desmin (1:50 unconcentrated hybridoma supernatant; D3, DHSB), Alexa Fluor-568 phalloidin (A12380, Invitrogen, 150 nM), and desmoglein 2 (ab150372, Abcam, 1:250), NaV1.5 (1:500; ASC-005, Alomone Labs), NaV1.5 rabbit sera (1:50, in house[4]) and purified β1 (1:500, in house[4]).

### Cardiac fibrosis analyses

Analyses of cardiac fibrosis in Tmem65 KD hearts were performed with ImageJ[65] as described below. Briefly, cardiac images after Masson's Trichrome staining were re-sized to reflect the physical dimensions of images. Images were converted to gray scale (*Image > Type > RGB Stack* command), threshold (*Image > Adjust > Threshold* command) established with the red channel, and fibrotic areas (*Analyze > Measure* command) measured by ImageJ.

### Echocardiography

A high frequency ultrasound imaging system (Vevo770, VisualSonics Inc., Toronto) with a 30 MHz transducer. Mice were anesthetized using isoflurane (induced at 5% in medical oxygen, and maintained at 1.5% through face mask). Mice were positioned supine with four paws taped to electrodes on a pre-warmed platform for heart rate monitoring. Mouse body temperature was monitored by rectal thermometer and maintained at 37 °C. Mouse hair on the whole chest was cleanly removed using hair-removal cream (Nair). The ultrasound imaging procedure was then performed[66]. Briefly, the left ventricle in its long axis view was visualized using two-dimensional imaging to observe its overall morphology. The M-mode trace recorded from the middle segment of the left ventricle was used to measure the left ventricular anterior and posterior wall thicknesses and chamber dimensions at peak-systole and end-diastole, and the left ventricular ejection fraction (EF%) and fractional shortening (FS%) were calculated. The diameter of aortic annulus and the Doppler flow velocity at the middle of aortic orifice were measured for calculating the left ventricular stroke volume and cardiac output.

## Electron microscopy

Mouse hearts were perfused with ice-cold 1x PBS and fixed in 2% paraformaldehyde at 4 °C for at least 2 days. After fixation, samples were cut from midmyocardium, washed in 0.1 M sodium cacodylate buffer (pH 7.4), treated with osmium tetroxide, dehydrated, infiltrated and embedded using increasing concentrations of Araldite/Epon 812. Tissue samples were sectioned to 75 nm thickness and stained with uranyl acetate followed by Reynold's Lead Citrate. Standard transmission electron microscopy was performed at the St. Michael's Hospital (Toronto, Ontario, Canada)[16]. Image analyses were performed using ImageJ analysis software (NIH). Perinexal images were collected using a JEOL 1400 Plus transmission electron microscope (EM) with an Advanced Microscopy Techniques side-mounted 8k bottom-mount digital camera at the University of South Carolina School of Medicine, Columbia, SC. Tissue samples from mouse ventricles were embedded, sectioned, and mounted on EM grids[4,36]. Sections were stained with Uranyless for 1 min, washed, and counterstained with Reynold's lead citrate solution for 7 min, then allowed to dry for at least 24 h before imaging. Sections were scanned at ×5000 magnification for ICDs in the EM. Once an ICD was identified, regions were then scanned visually for gap junctions and surrounding perinexus. Images of gap junctions and associated perinexus were taken at 40,000×. For analysis, all perinexus images were included based on the following requirements: directly adjacent to identifiable gap junction, clear separation of membranes at the perinexal cleft, measurable width up to 105 nm from the edge of the gap junction. The perinexus was manually traced in ImageJ and then analyzed with MATLAB[4,36,67]. Specifically, averages of perinexal width from 30–105 nm from the gap junction are the final reported values. Non-perinexal measurements were taken in ImageJ from the same images at distances greater than 200 nm from the gap junction edge. Statistical analysis was performed using a 2-tailed nested $t$-test between the scrambled control shRNA group ($n = 3$ animals, 15–20 perinexi per animal) and the Tmem65 shRNA group ($n = 4$ animals, 15–20 perinexi per animal). Perinexal and non-perinexal width measurements were found to conform to a Normal distribution (Kolmogorov-Smirnov test with Dallal–Wilkinson–Lilliefor $P$ value, $p > 0.05$) and showed uniform variance following log transformation. Thus, the nested $t$-test comparison between control and treatment was carried out using a log-transformed data set.

## Super-resolution microscopy

Fluorophore-antibody conjugation was prepared using standard procedures[68]. Briefly, 10 µg of each purified antibody were mixed together with dibenzocyclooctyne-PEG4-$N$-hydroxysuccinimidyl ester (Sigma, 764019), fluophore-$N$-hydroxysuccinimidyl ester (Invitrogen, Alexa Fluor 488-NHS, A20000; Alexa Fluor 633-NHS, A20005) in the molar ratio of 1:20:0.5. Conjugation reaction was completed within an hour at room temperature and unconjugated fluorophore was removed with micro Bio-Spin chromatography (Bio-Rad, 732–6200) according to manufacturer's instruction. Mouse cardiac sections were prepared as described above, but were bonded on 1-µm coverslips. Immunofluorescence was performed with fluorophore-conjugated antibodies (1:50 dilution) overnight at 4 °C. Two-color STORM imaging was performed on a modular imaging system built around an IX-83 base (Olympus, Canada) using a 100× oil-immersion TIRF objective (UApoN 100×/1.49 Oil, Olympus, Japan). Alexa Fluor 488 (AF488) was excited with a 488 nm laser (Sapphire 200 mW, Coherent, USA) with the laser power modulated to 20 mW at the objective. Similarly, Alex Fluor 633 (AF633) was excited with a 647 nm laser (200 mW, Laser Glow, Canada) with the laser power modulated to 20 mW at the objective. A detailed snapshot of the optical system is described in the following file (https://github.com/YipLab/IX83-Modules/releases/tag/1.1). For STORM imaging, 10,000 images over a period of 300 s were acquired each with an exposure of 30 ms on a Prime BSI (Photometrics, USA) and acquired sequentially for both colors. To facilitate photoswitching of the fluorophores an oxygen scavenging buffer (50 mM cysteamine, 2-mercaptoethylamine, 40 µg/ml catalase, 0.5 mg/ml glucose oxidase Sigma-Aldrich, 50% w/v glucose in PBS) was used. Image stacks were analyzed using the ThunderSTORM plugin in Fiji using the default settings[69]. The localized coordinates were then filtered based on localization precision (uncertainty value) to remove electronic noise (0 nm < localization precision < 5 nm) and sample noise (localization precision > 15 nm).

## Electrocardiographic (ECG) recordings in conscious mice

Mouse ECG was recorded and analyzed by ECGenie (Mouse Specifics Inc, Framingham, MA) per manufacturers' instructions[29]. Briefly, mice injected with virus were housed in a regular facility with 12-h light/12-h dark cycle for 3 weeks. Mice ECG was recorded between 8:30 and 11:30 in the morning to avoid the influences of circadian rhythm. An array of gel-coated AgCl ECG electrodes (MSI001A, Mouse specifics Inc., Boston, MA) were embedded in the floor of the platform to provide contact between the electrodes and animals' paws. The electrodes were connected to an amplifier (e-Mouse). ECG signals were digitized at a sampling rate of 2 kHz with 16-bit precision by Powerlab and LabChart (ADInstruments, version 7). For minimizing stress-induced irregularity, mice were positioned on the recording platform for 10-min acclimatization before at least 20-min recording. A peak detection algorithm on LabChart enabled R-wave identification using Fourier analysis and linear time-invariant digital filtering of frequencies 3 Hz (high pass), 100 Hz (low pass), and 60 Hz notch filtering. Segments of continuous recordings (15–20 P-Q-R-S-T complexes) were analyzed using the eMouse software (Mouse Specifics) that generated ECG parameters based on the standard default algorithm[29,70].

## Voltage measurements in isolated mouse hearts using voltage-sensitive dyes

According to our institutional animal care procedures[71], mice were injected with (200 IUs) heparin and 5 min later were anesthetized deeply with 4% isoflurane (in $O_2$). Following cervical dislocation, the thorax was opened by midsternal incision, and the hearts were excised and placed into ice-cold Kreb's solution containing: 118 mM NaCl, 4.2 mM KCl, 1.2 mM $KH_2PO_4$, 1.8 mM $CaCl_2$, 1.2 mM $MgSO_4$, 23 mM $NaHCO_3$, 20 mM D-glucose, and 2 mM sodium pyruvate (pH 7.4 adjusted via bubbling with a 95% $CO_2$/5% $O_2$ carbogen gas). The heart was rapidly mounted onto a horizontal Langendorff apparatus and the aorta was mounted onto a blunted 20-gauge needle connected via a water-jacketed perfusion line to perfusion pump with variable flow rates (2.0–2.5 ml/min) adjusted to maintain a perfusion pressure of 70–90 mmHg (Radnoti LLC, Covina, CA). The perfusion pump received heated Kreb's solution from a reservoir container (gassed with carbogen) and containing 20 mM 2, 3-butanedione monoxime (Sigma-Aldrich, B0753) to inhibit contractile protein force generation[72], thereby minimizing movement artefacts. The temperature of the perfused hearts was monitored constantly and maintained at 36 ± 1.0 °C throughout the experiment. Viable hearts regained a pink colouration and spontaneous rhythmic contraction with perfusion, with hearts excluded if sinus rate dropped below 250 bpm or if AV nodal block occurred.

After the heart were mounted and perfused, 3 Ag/AgCl electrodes were positioned within 1 mm of the heart in a lead II electrogram configuration. The electrodes were attached to a Biopac amplifier (UIM-100C, CA., USA) whose output was digitized (AXON CNS Minidigi 1B, Molecular Devices, CA., USA) and displayed continuously to assess spontaneous rhythmic activity. If sinus rate dropped below 250 bpm or if AV nodal block occurred, experiments were terminated immediately, and the heart was not used in subsequent analyses.

After perfusion for 15-following mounting, the heart was perfused with a "loading" solution consisting of Kreb's with 20 mM BDM plus voltage-sensitive dye, Di-4-ANEPPS (2 µM, Santa Cruz, sc-214872, USA). After 5–7 min of perfusion with the Di-4-ANEPPS loading solution,

hearts were perfused again with Kreb's solutions plus 20 mM BDM. The anterior epicardial surface was then excited using a high-powered LED illumination system (LEX2-LZ4, 530 nm peak wavelength). Light generated was controlled by an electronic shutter and then passed through a band-pass filter (531 ± 40 nm) and fluorescent light was passed through a 610 nm long-pass filter (Semrock, Rochester, NY). Images were collected from a 14 × 14 mm field of view using a 0.63× objective lens (NA = 0.35) and projected onto a complementary metal oxide semiconductor (CMOS) camera equipped with sensors containing 100×100 pixels (MiCAM Ultima-L, SciMedia, Costa Mesa CA., USA). Images were collected at frame rates of 1 kHz.

Optical recordings were made during sinus rhythm or were paced at a 90-ms cycle length by applying 1 ms pulses at a voltage 1.5x the capture threshold applied to the epicardial surface (between right ventricular free wall and apex) using platinum electrodes (spaced ~1 mm apart) attached to a stimulator (Pulsar i6 Stimulator, Frederick Haer & Co (FHC), Bowdoinham, ME). Images were stored using MiCAM Ultima Experiment Manager. Activation maps were generated using 4 s of continuous optical recordings which were used to calculate conduction velocity using the MATLAB-based ElectroMap electrophysiology mapping software[73].

## Current measurements using whole-cell patch-clamp recordings

Mouse adult ventricular myocytes were isolated from adult mice[74]. Briefly, CD1 mice were euthanized by open drop exposure to isoflurane followed by cervical transduction. The chest cavity was opened, the descending aorta severed, and 7 ml of EDTA buffer containing 15 μM blebbistatin (B592500, Toronto Research Chemicals, Toronto, ON) was injected into the right ventricle. The heart was hemostatically clamped at the ascending aorta, excised from the chest cavity, and placed into a dish of fresh EDTA buffer containing 15 μM blebbistatin while 9 ml of the same buffer was injected slowly into the apex of the left ventricle. After the heart was cleared of blood, it was moved to a dish of perfusion buffer with 15 μM blebbistatin, and injected with 3 ml of fresh 15 μM blebbistatin through the same hole previously used in the left ventricle. Finally, the heart was moved to a dish containing 475 U/ml collagenase type II (Worthington Biochemical Corporation, Lakewood NJ) in perfusion buffer with 15 μM blebbistatin, of which 20 ml more was injected through the existing left ventricle opening. We found that the use of 27 G, ½ length needles minimized mechanical damage to the heart, allowing for the maintenance of pressure during perfusion and optimal coronary circulation of collagenase and, thus, digestion of the myocardium. Following collagenase digestion, tissue was minced in 3 ml of fresh collagenase buffer with forceps, and gently triturated with a wide-bore 1 ml pipette. Collagenase activity was inhibited with the addition of 3 ml of perfusion buffer with blebbistatin and 10% FBS. The isolate was then passed through a 70 μm strainer and rinsed with 3 ml additional stop buffer. The filtrate was divided between two 15 ml Falcon tubes which were left standing upright for 15 min. The rod-shaped viable cardiomyocytes gravity-settled to form a deep red pellet, while rounded, nonviable adult cardiomyocytes and other cell types remained in suspension. The use of 2 tubes prevented oxygen or nutrient gradients forming in the cell pellets, while the use of steep-walled 15 ml Falcon tubes allowed for the best recovery of the pellet over successive washes. The supernatant was removed carefully, and the cells resuspended in a mixture of 75% perfusion buffer and 25% culture media, containing 15 μM blebbistatin. Cells were allowed to settle 15 min, and the process repeated two more times with mixtures of 50%:50% and 25%:75% perfusion buffer:culture media, respectively, all containing 15 μM blebbistatin. The final cell pellet was resuspended in culture medium containing 5% FBS and 15 μM blebbistatin.

Isolated cardiomyocytes were placed in perfusion chamber located on the stage of an IX70 microscope and continuously perfused with a Tyrode's solution (called a bath solution) containing: 140 mM NMDG, 5 mM NaCl, 4 mM KCl, 1 mM MgCl₂, 2 mM CaCl₂, 10 mM HEPES, 10 mM D-glucose, pH = 7.35 adjusted by HCl. Cells were imaged using a 40X objective (Olympus, Tokyo, UPlanApo, Japan) and cardiomyocytes with sharp edges and clear sarcomeric patterns were used to make giga seals with polished (Micro Forge MF-90, Narishige Group, Japan) filamented borosilicate glass pipettes (1.5 mm OD, 1.12 mm ID, World Precision Instruments, FL, USA) created using a pipette puller (p-87, Sutter Instrument Company). Pipette resistances were 3–6 MΩ after filling with intracellular solution. After a giga seal was formed and the membranes were ruptured to gain whole-cell access, membrane currents were measured in the voltage-Clamp configuration using the Axopatch 200B voltage-clamp amplifier (Molecular Devices, CA., USA) and digitized (Axon Digidata 1440A) using recorded using pClamp 10.7 (Molecular Devices, CA., USA). The current recordings are analyzed offline using Clampfit 10.7 (Molecular Devices, CA., USA). All recordings were performed at room temperature perfusing with bath solution.

For voltage-gated Na⁺ and voltage-gated Ca²⁺ currents, the Na⁺ pipette solutions contained (mM): 135 CsCl, 5 NaCl, 1 CaCl₂, 10 EGTA, 1 MgCl₂, 10 HEPES and 4 MgATP; (pH adjusted to 7.2 with CsOH). The bath solution consists of (mM): 140 N-methy-D-glucamine, 5 NaCl, 4 tetraethylammonium chloride, 1 MgCl₂, 2 CaCl₂, 10 HEPES, 10 glucose (pH adjusted to 7.35 with HCl). Prior to making voltage-clamp recordings, series resistance was compensated by 80–90%. Current recordings were filtered at 2 kHz and sampled at 50 kHz. Voltage-gated Na⁺ plus Ca²⁺ currents were quantified as follows. First, we applied protocol 1 which consisted of 500 ms voltage steps between −70 mV and +50 mV in 5 mV increments to be applied from a holding voltage of −100 mV. The time between step recordings was 5 sec. Thereafter, protocol 2 was applied which consisted of the introduction of a 50 ms prepulse to −40 mV from a holding potential of −80 mV prior to the application of the same 500 ms voltage steps as protocol 1. The Na⁺ currents were estimated by subtracting the current traces measured in protocol 2 from the traces measured in protocol 1 while Ca²⁺ current was determined from protocol 2. The conductance ($G_X$) was determined using the formula: $G_X = (I_{peak}/(V_m − E_{rev}))/G_{max}$, where "X" represents either Na⁺ or Ca²⁺, $I_{peak}$ is the peak inward current during steps to $V_m$, $E_{rev}$ is the measured reversal potential for Na⁺ or Ca²⁺ current and $G_{max}$ in the maximum conductance measured. $G_X$ was fit to the Boltzmann function (GraphPad Prism software, V9.02) in order to estimate $V_{1/2}$ (the voltage for 50% activation) and slope factor (i.e., an estimate of the gating charge); differences in $V_{1/2}$ and slope factors were compared using extra sum-of-squares F test.

The voltage-dependence of inactivation of Na⁺ and Ca²⁺ channels was determined by applying 500 ms prepulse voltage steps between −100 mV and +20 mV (5 mV increments), from a holding potential of −120 mV, followed by a test-pulses to −30 mV for 30 ms (to quickly activate Na⁺ channels) and a subsequent test pulse to +10 mV for 300 ms (to activate Ca²⁺ channels). Steady-state inactivation curves were generated by plotting $I_{peak}/I_{max}$ as a function of the prepulse voltage where $I_{peak}$ is the peak current measured during the test pulse and $I_{max}$ is the maximum value of $I_{peak}$ (measured at prepulses to −120 mV). Inactivation curves were also fit to the Boltzmann function in order to estimate the $V_{1/2}$ for Na⁺ channel inactivation and slope factor; differences in $V_{1/2}$ and slope factors were compared using extra sum-of-squares F test.

For the K⁺ currents recordings and estimations, cells were perfused with bath solution containing (mM): (140 NaCl, 4 KCl, 1 MgCl₂, 1.2 CaCl₂, 10 HEPES, 0.3 CdCl₂, 10 glucose (pH 7.4 with NaOH)). The pipette was filled with a solution containing (mM): 120 potassium aspartate, 20 KCl, 5 NaCl, 1 MgCl₂, 5 MgATP, 10 HEPES, and 10 EGTA (pH 7.2 with KOH). All chemical reagents were obtained from Sigma-Aldrich. A voltage-clamp protocol (holding at −80 mV, followed by prepulses to −40 mV for 30 ms and a subsequent step to +60 mV for 25 s) was applied.

The decay of the outward K⁺ current was fit nonlinearly to a triexponential function with a baseline offset in order to dissect the 4 different kinetic voltage-gated K⁺ components (i.e. the transient

outward K$^+$ current ($I_{to}$), the "fast" delayed rectifier current ($I_{K,Slow1}$ encoded by Kv1.5), the "slow" delayed rectifier current ($I_{K,Slow2}$) and the sustained current ($I_{K,SS}$)[31]. To separate the two components of I$_{to}$ (i.e. fast $I_{to}$ called $I_{to,f}$ encoded by Kv4.2/3 and slow $I_{to}$ called I$_{to,s}$ encoded by Kv1.4), we reapplied the same protocol after the application of 4-amino-pyridine (4-AP), at a dose (500 μM) that potently inhibit Kv1.4 ($I_{to,s}$) and Kv1.5 ($I_{K,Slow1}$) but not the other K$^+$ currents in mouse myocardium[31]. Consequently, bi-exponential fits to the 4-AP-sensitive current (obtained by subtracting outward K$^+$ currents measured in the presence of 4-AP from currents measured without 4-AP), allows an estimation of $I_{to,s}$ (as well as an additional estimation of $I_{K,Slow1}$). $I_{to,f}$ was then obtained by the difference between I$_{to}$ and $I_{to,s}$.

## Cloning of Scn1β
cDNA of Scn1β was amplified from mouse cardiac cDNA library by Q5 High-Fidelity 2x Master Mix (New England Biolab, M0492S). Briefly, 500 ng of mouse cardiac cDNA and 10 μM each primer (5′-ATA TGG TAC CAT GGG GAC GCT GCT GGC T-3′, Scn1β SP, 5′-ATA TGG ATC CCT ATT CAG CCA CCT GGA CGC C-3′; Scn1β AP) were mixed per manufacturer's instruction. PCR cycling was carried out as the following, initial denaturation at 98 °C for 30 s, 40 cycles of 98 °C for 10 s, 60 °C for 10 s for primer annealing, 25 s for polymerization, and final extension at 72 °C for 2 min. PCR amplicons were then subcloned to pcDNA3 plasmids for generating a pcDNA3-Scn1β expression plasmid, which was confirmed by Sanger sequencing at ACGT Corp (Toronto, ON, Canada).

## Statistical analyses
All data are expressed as mean ± standard error of the means (SEM). Statistical analyses were performed by one-way ANOVA with Tukey's post-hoc test. For the image analysis of the EM images, nested *t*-test comparisons between control and treatment were carried out using the log-transformed data set. For optical mapping, data were expressed as mean ± SEM. Differences between scrambled shRNA and Tmem65 shRNA groups at sinus rhythm and during pacing were assessed with a two-way ANOVA with Sidak's multiple comparison test. For patch-clamp experiments, data were presented as mean ± standard deviation (SD). The difference of current density was compared using unpaired student *t*-test.

## Reporting summary
Further information on research design is available in the Nature Research Reporting Summary linked to this article.

## Data availability
Source data are provided with this paper.

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

## Acknowledgements

We thank Ms. Wenping Li and Mr. Aaron Wilson for their expert technical support in viral packaging and purification, and Ms. Venus Chan for assisting animal work. We thank Dr. Bob Price and Mr. Jeff Davis in the ICRF core at the Medical University of South Carolina Medical School for their assistance with electron microscopy. This project was funded by the Ted Rogers Centre for Heart Research Innovation Fund; the Heart and Stroke Richard Lewar Centre of Excellence in Cardiovascular Research, NIH (RGG-HL56728, HL141855, HL161237) and CIHR Grants (AOG-PJT155921, MOP106538, MOP123320, GPG102166, PJT149011; PHB-MOP125950). Canada Research Chair in Cardiovascular Biology to P.H.B.; the Canadian Foundation for Innovation, John Evans Leader Award to P.H.B. R.L. was supported by Canadian Institute of Health Research Postdoctoral Fellowship. A.C.T.T. was supported by Ted Rogers Centre for a Heart Research Fellowship.

## Author contributions

A.C.T.T, L.G., C.M.Y., R.M.H., R.G.G., C.A.S., P.H.B., T.K., and A.O.G. conceived study designs. A.C.T.T., L.G., M.D., R.L., Z.J.W., A.A., W.C., N.I.C., F.H.Z., Y.Z., M.F., D.C., L.J.J., and J.L. performed experiments and data analyses. A.C.T.T. and A.O.G prepared the initial manuscript draft. A.C.T.T., M.D., R.L., Z.J.W., M.F., C.M.Y., R.G.G., P.H.B., and A.O.G. performed extensive experimental or design revisions to the manuscript. All authors reviewed, edited and approved the final version of the manuscript.

## Competing interests

The authors declare no competing interests.

## Additional information

Allen C. T. Teng[1,2] ✉, Liyang Gu[2,13], Michelle Di Paola[1,2,13], Robert Lakin[3,13], Zachary J. Williams[4,5], Aaron Au [6,7], Wenliang Chen[3], Neal I. Callaghan [2,6], Farigol Hakem Zadeh[1,2], Yu-Qing Zhou[2,6], Meena Fatah[8], Diptendu Chatterjee[8], L. Jane Jourdan[4,9,10], Jack Liu[1], Craig A. Simmons [2,6], Thomas Kislinger[11,12], Christopher M. Yip [6,7], Peter H. Backx[3], Robert G. Gourdie [4,9,10], Robert M. Hamilton [8] & Anthony O. Gramolini [1,2] ✉

[1]Department of Physiology, Temerty Faculty of Medicine, University of Toronto, Toronto, ON M5S 1A8, Canada. [2]Translational Biology and Engineering Program, Ted Rogers Centre for Heart Research, Toronto, ON M5G 1M1, Canada. [3]Department of Biology, York University, Toronto, ON M3J 1P3, Canada. [4]The Center for Heart and Reparative Medicine, Fralin Biomedical Research Institute at Virginia Tech. Carilion, Roanoke, VA 24016, USA. [5]Translational Biology Medicine and Health Graduate Program, Virginia Tech, Roanoke, VA 24016, USA. [6]Institute of Biomedical Engineering, Faculty of Applied Science and Engineering, University of Toronto, Toronto, ON M5S 3G9, Canada. [7]Donnelly Centre, University of Toronto, Toronto, ON M5S 3E1, Canada. [8]The Labatt Family Heart Centre (Dept. of Pediatrics) and Translational Medicine, The Hospital for Sick Children & Research Institute, University of Toronto, Toronto, ON. M5G 1X8, Canada. [9]Virginia Tech Carilion School of Medicine, Roanoke, VA 24016, USA. [10]Department of Biomedical Engineering and Mechanics, Virginia Polytechnic Institute and State University, Blacksburg, VA 24060, USA. [11]Princess Margaret Cancer Centre, University Health Network, Toronto, ON M5G 1L7, Canada. [12]Department of Medical Biophysics, University of Toronto, Toronto, ON M5G 1L7, Canada. [13]These authors contributed equally: Liyang Gu, Michelle Di Paola, Robert Lakin. ✉e-mail: Allen.Teng@utoronto.ca; Anthony.Gramolini@utoronto.ca

