## [Peer Review File · Nature Communications]

REVIEWER COMMENTS

Reviewer #1 (Remarks to the Author):

The main concerns in this paper are:

- 1) There seems to be an underlying assumption that Nav1.5 is mostly localized to the so-called perinexus. This is mistaken. There is abundant Nav1.5 protein distributed along the sarcomeres, as demonstrated by a number of studies. Moreover, at the intercalated disc, a large fraction of the Nav1.5 protein localizes in a complex with N-Cadherin. Only a fraction of the total Nav1.5 localizes to the perinexus. Whether loss of this specific fraction (the other ones being intact) is sufficient to impair propagation, remains unclear.
- 2) There are multiple reasons that can lead to an arrhythmogenic and cardiomyopathic heart. Here there is consideration of only one variable, which is not even studied at the single cell electrophysiology level. Furthermore,
- 3) The q images shown in the manuscript are way below standard of quality.

This paper presents some initial suggestions as to how Tmem65 may affect cardiac function, but it is far from providing conclusive evidence in support of a firm hypothesis. Some more detailed comments:

What is the extent of AAV infection in the hearts? Which percentage of cells were infected? Though the Kaplan-Meier clearly shows that the injection is having an effect, documentation on the actual infection (*vis a vis*, say, inflammatory response) would be helpful. Moreover, though AAV9s are preferentially cardiac, the investigators should document that only (or mostly) the heart is infected. Could Tmem65 silencing have an effect on the liver, for example? Or in the lungs?

Line 264: are the authors suggesting that reduced Cx43 abundance causes decreased Gja1 expression? Is there any evidence in favor of that?

Line 266: how do the authors define "Cx43 puncta"? I do see less signal for Cx43 and for N-Cad but at the magnification used, it is not possible to see puncta. In that Figure, N-Cad staining in the shRNA frame is very weak. There are multiple nuclei apparent in the image, and only a few spots of N-Cad localization. The lower Pearson coefficient may be because of poor N-Cad staining.

EM of the ID: Using 2D EM to analyze the ID structure is too limiting, since the ID is a three-dimensional object. Modern technology allows for visualization in 3D. The image shown for the control shows only a couple of "ruffles" of the ID contour, but a more complex morphology may be present in other areas. The section on the right does not seem to be cut at the same angle as the one on the left, or the center. These images are difficult to interpret.

Line 374: I don't think the data are of enough resolution to assess the adhesion of Cx43 to the ID... Maybe the author was referring to the localization?

Fig.5C: the images seem a bit out of focus. These sections could be counterstained with alpha-actinin, so that one can evaluate the blurry staining versus something that is sharp and well defined.

Line 379-380: Why would the authors assume that loss of desmosomes would lead to disorganization of desmin at the z disks?

Fig.5d: These samples should be counterstained with something that does localize to the ID so that the authors can make a claim about the localization of desmosomal proteins in relation to the ID. Also, a marker of the sarcomeres would be helpful so that the reader can judge if the samples are cut at the same plane/angle.

Line 383: I do not think that the data provided allow the investigators to make such a claim.

Fig.6: Nav1.5 staining: it is well established that, when the total Nav1.5 is taken into account, the majority of it is localized not to the perinexus, but to the lateral membrane and to the ID in association with N-Cad. The control images show some staining at what may be a cell end, but no staining along the cell body. Western blots should show not only the little window but the entire lane, since Nav1.5 often degrades during the preparation and a big band appears at a lower mw.

Reviewer #2 (Remarks to the Author):

In this study, the authors probe the role of transmembrane protein 65 (Tmem65) in cardiac intercalated disc (ICD) structure and function. Tmem65 loss of function (LOF) appeared to be efficiently achieved in mice that were targeted with an adeno-associated virus 9 (rAAV9) expressing Tmem65 shRNA. Tmem65 shRNA mice exhibited heart failure-like symptoms in two phases of mortality, 3 and 7 weeks post viral administration, developing severe cardiac fibrosis, depressed ventricular hemodynamics and impaired conduction. Immunoprecipitation indicate an interaction between Tmem65 and sodium channel subunit $\beta 1$ (Scn1 β) and there was evidence of $\beta 1$, Nav1.5, CX43, and desmosomal loss of function in the Tmem65 targeted mice. The authors conclude that disruption of Tmem65 function is associated with impaired ICD and myofibril structure, ultimately leading to cardiomyopathy in vivo. The study appears well performed and its specific insights and discussion are of interest. The authors seem to suggest that the ephaptic nanodomains determined by $\beta 1$ are disrupted by Tmem65 LOF. There is some data provided for this, however as described in specific comments, further data may be required to support these claims.

Specific Comments:

Figure 1: The authors suggest that internalized Cx43-positive puncta were seen in Tmem65 KD mouse hearts. It may be difficult to distinguish lateralized from internalized Cx43 using this type of immunofluorescence (IF) approaches. If the authors want to claim that the Cx43 is internalized, further work may be required. The most direct demonstration would be in the EM, with a quantifiable increase in endocytosed annular GJs (connexosomes). Alternately, the authors could use co-localization of Cx43 with a cytoplasmic marker e.g., nuclear, lysosomal and confocal optical sectioning or super resolution microscopy to ensure the z plane of the section is in the cytoplasm. Also, higher magnification views should be provided– it is difficult to make out the signal on the current figure.

Also, in Figure 1 – the Westerns suggest some variability in down-regulation of Tmem65 and Cx43. What is the authors account of this ?

Figure 5 – It is useful to show EM of intercalated disks – as the authors have done. However, in light of subsequent data and discussions on GJ-associated $\beta 1$ further data is required. The should investigate and illustrate perinexal domains at the edge of GJs. If the authors hypothesis is correct Tmem65 and $\beta 1$ LOF should result in perinexal intermembrane widening – consistent with disruption of the ephaptic nanodomain ? A further and probably required step would be to document the extent of widening of the perinexi in control and rhRNA targeted mice to determine whether or not perinexal intermembrane spacing changes have occurred – especially any changes in this spacing that exceed the 30 nm distance required for ephaptic conduction to operate e.g., see figure 4 PMID: 30106376. The authors apparently have the necessary EM section in hand, so this task should not be too much of a reach. The lack of EM of the perinexus is the reviewer’s major concern.

Figure 6 - The Nav1.5 IF seems high in controls with complete loss in the ICD of treated mice, relative to Westerns. What is the authors explanation of this difference ? Also, what about Cx43 based interactions with sodium channel subunits. A number of groups have reported Cx43-Nav1.5 interact in the perinexus. Did the authors looks at this and if so were there alterations to this association with Tmem65 LOF?

Figure 7 The Ip data is convincing and well performed. However, have a further useful demonstration would be to show the interaction in situ. For example, use of a duolink polymerization ligation to directly localize Tmem65- β 1 associations at or near Cx43 GJs would be extremely persuasive. For that matter it would be useful and probably necessary to simply do and illustrate high magnification confocal (or super-resolution if available) of Cx43 positive GJ plaques to assess localization of Tmem65 at these structures. One of the best ways to do this is by imaging intercalated disks in en-face orientation as was also done in PMID: 30106376 see Figure 1.

Minor

Introduction Line 90-92: " Conversely, Cx43 trafficking and localization to the ICD depends on desmosomes 12 and tight junction proteins 8 possibly through at specialized ICD region called the perinexus."

Probably best not to invoke tight junctions - cardiomyocytes do not have these junctional structures

Reviewer #3 (Remarks to the Author):

Teng et al. report their results on AAV-mediated expression of a sh-RNA for knockdown of Tmem65 in neonatal mice. Mice developed cardiac hypertrophy and conduction abnormalities 3 weeks after AAV-shRNA application and dilated cardiomyopathy 7 weeks after AAV injection. Mechanistically, Tmem65 knock-down resulted in loss of ephaptic junctions at the intercalated disc. Overall, the results are novel and help to understand the role of Tmem65 with intercalated discs. Nevertheless, a number of issues needs further clarification.

Major:

- 1. The potential role of the gender is an interesting finding. However, the role of the gender for the findings reported in this manuscript are not clear. Could gender have played a role in the mice having died after intramyocardial injections? Has gender been determined in these mice or control mice? Overall, considering a gender effect, it would be helpful to specify gender information in all figure legends and tables.**
- 2. Animal numbers are not clearly indicated for all experiments. How many mice injected to investigate the early and late time point (figure 4, 5, and 6 or figure 2 and 3)? I did not find information on animal numbers (or gender) used for distinct measurements in the figure legends (only in the tables).**
- 3. It is difficult to spot differences in Scn1beta immunofluorescence in in Fig. 6e.**
- 4. Analysis of fibrosis on the histological level (Masson Trichrome or Sirius red stains) would be helpful.**
- 5. Please provide details on the shRNA sequence. Was it expressed within a microRNA context to be suitable for expression with the chicken troponin T promoter?**

Minor:

Line 46: Introduce „KD“ in the abstract.

Line 50: Delete the comma.

Line 102: grammar – rather „interact“`?

Line 110: Provide reference.

Line 124: Please cite the correct source of pDG9.

Figure 8: It is difficult to understand that the left part is the normal situation (scrambled shRNA) while the right part of the sketch is the Tmem65 KD situation. Please revise the otherwise helpful figure.

Reviewer #1 (Remarks to the Author):

The main concerns in this paper are:

- 1) There seems to be an underlying assumption that Nav1.5 is mostly localized to the so-called perinexus. This is mistaken. There is abundant Nav1.5 protein distributed along the sarcomeres, as demonstrated by a number of studies. Moreover, at the intercalated disc, a large fraction of the Nav1.5 protein localizes in a complex with N-Cadherin. Only a fraction of the total Nav1.5 localizes to the perinexus. Whether loss of this specific fraction (the other ones being intact) is sufficient to impair propagation, remains unclear.

Response: We are grateful for the reviewer's comment highlighting that there is a wide distribution of Nav1.5 in mammalian cardiomyocytes; and we obviously agree that there is localization of Nav1.5 across the entire sarcolemmal. In this study, we strategically focused on the voltage-gated sodium channel Nav1.5 in the intercalated discs (ICDs) for the following reasons. First, we show in the revised manuscript that Nav1.5 remained present in the cardiac sarcolemma in Tmem65 KD mouse cardiomyocytes (Supp. Fig. 4). In comparison, Nav1.5 presence in ICDs was evidently lost in Tmem65 KD cardiomyocytes (Fig. 6e). These results support our model that Nav1.5 localization to the ICD is partially mediated by Tmem65 in cardiomyocytes. This finding is consistent and supported by our new cardiac action potential studies – there were no significant differences found in the cardiac action potential duration between Tmem65 KD and control cardiomyocytes (Figs 6g & 6h). A similar observation was made recently in our previous publication (Veeraraghavan et al., 2018, Elife, PMID 30106376) that perturbed activity of the ICD-bound Nav1.5 did not affect cardiac action potentials in adult mouse cardiomyocytes. Instead, reduced ICD Nav1.5 activity was associated with reduced conduction velocity. Consistently, our previous study also reported both a significantly reduced conduction velocity and unaffected action potential in mouse neonatal cardiomyocytes following Tmem65 silencing (Sharma et al., 2015, Nat. Commun., PMID 26403541). The findings from our group and others' support that notion that Tmem65 is critical for Nav1.5 localization to the ICD in mouse cardiomyocytes and this statement does not exclude NCad-bound Nav1.5. Finally, whether the loss of Tmem65 would have impact on the sarcolemmal Nav1.5 remains unclear and is not evident by our study. In our revised version, we have ensured that the rational and experimental limitations described above are clearly communicated in the revised text (Pages 29-30, lines 601-628).

- 2) There are multiple reasons that can lead to an arrhythmogenic and cardiomyopathic heart. Here there is consideration of only one variable, which is not even studied at the single cell electrophysiology level. Furthermore,

Response: We agree with the reviewer that the action potential of isolated cardiomyocytes would provide significant improvement of the manuscript. As a result, we have revised our manuscript carefully, and have now included new experiments of both isolated cell action potential analyses as well as Ca²⁺ transients studies in isolated cells (Revised Fig 6, Supp. Fig 5, Tables 4 and 5). Interestingly, we found that there is

no difference in the action potential between control and Tmem65 KD cardiomyocytes (Figs. 6g & 6h). As mentioned above, disrupted ICD Nav1.5 activity did not affect overall myocyte action potential under physiological condition and this is consistent with our previous report (Veeraraghavan et al., 2018, Elife, PMID 30106376). In that study, we also found a decreased conduction velocity in affected cardiomyocytes, and our previous study (Sharma et al., 2015, Nat. Commun., PMID 26403541) showed an identical outcome in mouse neonatal cardiomyocytes in response to Tmem65 silencing. Together, these studies begin revealing the role of ephaptic junctions at the ICDs and our studies show Tmem65 is an important contributor to ephaptic junctions. These findings are included (Page 24, lines 510-522) and discussed in the revised manuscript (Pages 29-30, lines 601-628).

3) The q images shown in the manuscript are way below standard of quality.

Response: We appreciate the reviewer's comment concerning image quality of submission. We now provide new confocal images, and ensured that all image quality has been considerably improved in the revised manuscript. (Figs. 1e, 5a, 5c, Supp. 1b.)

This paper presents some initial suggestions as to how Tmem65 may affect cardiac function, but it is far from providing conclusive evidence in support of a firm hypothesis. Some more detailed comments:

What is the extent of AAV infection in the hearts? Which percentage of cells were infected? Though the Kaplan-Meier clearly shows that the injection is having an effect, documentation on the actual infection (vis a vis, say, inflammatory response) would be helpful. Moreover, though AAV9s are preferentially cardiac, the investigators should document that only (or mostly) the heart is infected. Could Tmem65 silencing have an effect on the liver, for example? Or in the lungs?

Response: AAV infectivity across the whole animal is always something to be concerned about, and we agree with the reviewer. The safety, efficiency, and low immunogenicity of AAV9-mediated shRNA delivery to murine hearts have been systematically established and reported by different laboratories (Prasad et al., 2010, Gene Therapy, PMID 20703310; Zacchigna et al., 2014 Circ. Res. PMID 24855205; Zaleta-Rivera et al., 2019, Circ. PMID 31315475) that we ensure are referenced in the revised manuscript (Page 6, lines 123-132; Page 27, lines 555-575). The amount of virus and delivery method in our study are identical to the study reported by Prasad et al. (PMID 24086659), which shows a high degree of cardiac specificity, but not detectable in the liver. The extent of viral infection in the heart follows the Poisson distribution (Prasad et al., 2010, Gene Therapy, PMID 20703310) and corresponds to 20 viral genomes per cells in our study. In addition, viral infection- and injection-related complications, such as cardiac inflammation, had been closely monitored throughout the study, but were never found in both scrambled control and Tmem65 knockdown mice. We have included this important information for clarification and proper documentation of experimental details in the manuscript as suggested (Page 27, lines 555-575).

Line 264: are the authors suggesting that reduced Cx43 abundance causes decreased Gjal expression? Is there any evidence in favor of that?

Response: This is a good point. No, there is currently no evidence to suggest such mechanism and our initial sentence was an over-reach. Instead, we attribute reduced Cx43 proteins to a reduced Gjal transcription expression in Tmem65 knockdown hearts. This sentence has been rewritten for clarification. (Page 17, lines 345-349).

Line 266: how do the authors define “Cx43 puncta”? I do see less signal for Cx43 and for N-Cad but at the magnification used, it is not possible to see puncta. In that Figure, N-Cad staining in the shRNA frame is very weak. There are multiple nuclei apparent in the image, and only a few spots of N-Cad localization. The lower Pearson coefficient may be because of poor N-Cad staining.

Response: We apologize for any confusion in our initial manuscript. We have improved the quality of presented images which now allow for a much clearer visualization of Cx43 in cardiac tissues (Figures 1e and f). We remove the word puncta as it was vaguely defined. Instead, we only discussed internalized Cx43 in the revised manuscript. In addition, we also studied the location of these internalized Cx43 and found a significantly increased colocalization between Cx43 and a mitochondrial marker Cox4 in Tmem65 KD mouse hearts (Figure 1g and h). Our findings demonstrate that Cx43 relocates to cardiac mitochondria, which may have an unknown function in response to reduced Tmem65 protein levels. The new findings and discussion are now included in the manuscript (Page 17, lines 354-358).

EM of the ID: Using 2D EM to analyze the ID structure is too limiting, since the ID is a three-dimensional object. Modern technology allows for visualization in 3D. The image shown for the control shows only a couple of “ruffles” of the ID contour, but a more complex morphology may be present in other areas. The section on the right does not seem to be cut at the same angle as the one on the left, or the center. These images are difficult to interpret.

Response: We appreciate this constructive input from the reviewers. We have performed additional experiments using both high resolution transmission electron microscopy and super resolution imaging of the cardiac cell and include these studies in the revised manuscript (Figs. 5a and 7a). These new studies support our findings and highlight the altered ICD in TMEM65 knockdown.

Line 374: I don't think the data are of enough resolution to assess the adhesion of Cx43 to the ID... Maybe the author was referring to the localization?

Response: We appreciate review's suggestion. See above. We have added in high quality images for a better assessment of Cx43 localization to the intercalated discs, indicative of internalized Cx43 protein signals (Fig. 1e and Supp. Fig. 1b). This is consistent with reviewer's suggestion and the manuscript has been revised to reflect the observation (Page 17, lines 354-358).

Fig.5C: the images seem a bit out of focus. These sections could be counterstained with alpha-actinin, so that one can evaluate the blurry staining versus something that is sharp and well defined.

Response: We appreciate reviewer's suggestion. We have ensured that higher quality images have been added for an accurate assessment. (Fig. 5c).

Line 379-380: Why would the authors assume that loss of desmosomes would lead to disorganization of desmin at the z disks?

Response: We thank the reviewer for this critical comment. To address this issue, we studied the sarcomeric desmin patterns via high quality immunofluorescence. Our results show that sarcomeric desmin is equivocally disrupted in Tmem65 KD within cardiomyocytes, while it remains intact in scrambled KD samples (Fig. 5C). These results imply that the structure of sarcomeric desmin is likely affected as a secondary effect of Tmem65 silencing in cardiomyocytes. Since the underlying mechanism remains unclear, we carefully state that future studies will be needed for understanding how the reduced Tmem65 function would gradually affect sarcomeric desmin structure (Page 22, lines 462-466; Page 28, lines 581-598).

Fig.5d: These samples should be counterstained with something that does localize to the ID so that the authors can make a claim about the localization of desmosomal proteins in relation to the ID. Also, a marker of the sarcomeres would be helpful so that the reader can judge if the samples are cut at the same plane/angle.

Response: We appreciate the suggestion and counter staining experiments have now been including N-Cad and these images are now added (Figs. 5d and 5e).

Line 383: I do not think that the data provided allow the investigators to make such a claim.

Response: The claim has been modified accordingly. (Page 22, line 469-471)

Fig.6: Nav1.5 staining: it is well established that, when the total Nav1.5 is taken into account, the majority of it is localized not to the perinexus, but to the lateral membrane and to the ID in association with N-Cad. The control images show some staining at what may be a cell end, but no staining along the cell body. Western blots should show not only the little window but the entire lane, since Nav1.5 often degrades during the preparation and a big band appears at a lower mw.

Response: We appreciate reviewer's comment on a wide distribution of Nav1.5 staining pattern in cardiomyocytes. We have detected Nav1.5 on both sarcolemma and intercalated discs in cardiac sections throughout this study. In the revised manuscript, both sarcolemmal and ICD Nav1.5 are included for clarification (Fig. 6e and Supp. Fig. 4). In addition, a full western blot showing Nav1.5 protein levels in Tmem65 KD hearts is also included for clarity (Fig. 6d).

Reviewer #2 (Remarks to the Author):

In this study, the authors probe the role of transmembrane protein 65 (Tmem65) in cardiac intercalated disc (ICD) structure and function. Tmem65 loss of function (LOF) appeared to be efficiently achieved in mice that were targeted with an adeno-associated virus 9 (rAAV9) expressing Tmem65 shRNA. Tmem65 shRNA mice exhibited heart failure-like symptoms in two phases of mortality, 3 and 7 weeks post viral administration, developing severe cardiac fibrosis, depressed ventricular hemodynamics and impaired conduction. Immunoprecipitation indicate an interaction between Tmem65 and sodium channel subunit $\beta 1$ (Scn1 β) and there was evidence of $\beta 1$, Nav1.5, CX43, and desmosomal loss of function in the Tmem65 targeted mice. The authors conclude that disruption of Tmem65 function is associated with impaired ICD and myofibril structure, ultimately leading to cardiomyopathy in vivo. The study appears well performed and its specific insights and discussion are of interest. The authors seem to suggest that the ephaptic nanodomains determined by $\beta 1$ are disrupted by Tmem65 LOF. There is some data provided for this, however as described in specific comments, further data may be required to support these claims.

Specific Comments:

Figure 1: The authors suggest that internalized Cx43-positive puncta were seen in Tmem65 KD mouse hearts. It may be difficult to distinguish lateralized from internalized Cx43 using this type of immunofluorescence (IF) approaches. If the authors want to claim that the Cx43 is internalized, further work may be required. The most direct demonstration would be in the EM, with a quantifiable increase in endocytosed annular GJs (connexosomes). Alternately, the authors could use co-localization of Cx43 with a cytoplasmic marker e.g., nuclear, lysosomal and confocal optical sectioning or super resolution microscopy to ensure the z plane of the section is in the cytoplasm. Also, higher magnification views should be provided– it is difficult to make out the signal on the current figure.

Response: We thank the reviewer for suggesting an alternative idea for assessing Cx43 internalization. Our recent findings show that internalized Cx43 co-localized with mitochondrial marker Cytochrome c oxidase subunit 4 (Cox4) in adult mouse cardiomyocytes in response to Tmem65 silencing (Figs 1g and h). In addition, higher magnification images are now added to the revised manuscript for a better presentation (Supp. Fig. 1b).

Also, in Figure 1 – the Westerns suggest some variability in down-regulation of Tmem65 and Cx43. What is the authors account of this?

Response: We agree that there is some variation in the amount of Tmem65 and Cx43 proteins on immunoblots. Unlike genetic knockout, AAV9-mediated transgene delivery in hearts is more complex with variation due to viral infection, response times following injection, and the animals immune responses. However, our results show a rather consistent reduction in mRNA and protein for both Tmem65 and Cx43 following Tmem65 KD. The concern over protein variation and clarification are included in the discussion for further clarification (Page 27, lines 554-563).

Figure 5 – It is useful to show EM of intercalated disks – as the authors have done. However, in light of subsequent data and discussions on GJ-associated $\beta 1$ further data is required. The should investigate and illustrate perinexal domains at the edge of GJs. If the authors hypothesis is correct Tmem65 and $\beta 1$ LOF should result in perinexal intermembrane widening – consistent with disruption of the ephaptic nanodomain? A further and probably required step would be to document the extent of widening of the perinexi in control and shRNA targeted mice to determine whether or not perinexal intermembrane spacing changes have occurred – especially any changes in this spacing that exceed the 30 nm distance required for ephaptic conduction to operate e.g., see figure 4 PMID: 30106376. The authors apparently have the necessary EM section in hand, so this task should not be too much of a reach. The lack of EM of the perinexus is the reviewer's major concern.

Response: We appreciate reviewer's suggestion for further improving the quality of this study. Images and analyses from transmission electron microscopy over ephaptic nanodomains have been investigated in the revised manuscript. Figures 7d and 7e shows a significant increase in perinexal intermembrane distance following Tmem65 silencing. Morphological increases of perinexal domains are evidently associated with impaired $\beta 1$ function at the ICD (Veeraraghavan et al, 2018, Elife, PMID 30106376), which is consistent with the loss of ICD $\beta 1$ in cardiomyocytes following Tmem65 silencing (Fig. 6g). Together, these studies demonstrate that Tmem65 plays a critical role in maintaining the ICD perinexus in mouse hearts along with $\beta 1$. The manuscript is revised to include this exciting finding (Pages 25-26, lines 543-550).

Figure 6 - The Nav1.5 IF seems high in controls with complete loss in the ICD of treated mice, relative to Westerns. What is the authors explanation of this difference? Also, what about Cx43 based interactions with sodium channel subunits. A number of groups have reported Cx43-Nav1.5 interact in the perinexus. Did the authors looks at this and if so were there alterations to this association with Tmem65 LOF?

Response: Nav1.5 proteins are largely found in sarcolemma and intercalated discs of mammalian cardiomyocytes. In the revised manuscript, we show a significant amount of sarcolemmal Nav1.5 in both control and Tmem65 KD hearts (Supp. Fig. 4). These data suggest loss of Tmem65 impact mostly Nav1.5 localization to the ICD. This line of evidence is further supported by electrophysiology studies. Specifically, cardiac action potential did not differ in control and Tmem65 KD cardiomyocytes (Figs. 6g and 6h). This is consistent with our earlier work (Veeraraghavan et al., 2018, Elife, PMID 30106376) that disrupted Nav1.5 activity at the ICD never affects action potential in cardiomyocytes. Instead, there is a reduced conduction velocity, a similar observation we previously reported in mouse neonatal cardiomyocytes following Tmem65 KD (Sharma et al., 2015, Nat. Commun., PMID 26403541). Also, Tmem65 LOF is likely associated with reduced Cx43 and Nav1.5 interaction in cardiomyocytes. This is particularly true due to both reduced protein levels (Figs 1c and 6f) and a decreased ICD localization (Figs. 1e and 6g) in response to Tmem65 silencing.

Figure 7 The Ip data is convincing and well performed. However, have a further useful demonstration would be to show the interaction in situ. For example, use of a duolink

polymerization ligation to directly localize Tmem65- β 1 associations at or near Cx43 GJs would be extremely persuasive. For that matter it would be useful and probably necessary to simply do and illustrate high magnification confocal (or super-resolution if available) of Cx43 positive GJ plaques to assess localization of Tmem65 at these structures. One of the best ways to do this is by imaging intercalated disks in en-face orientation as was also done in PMID: 30106376 see Figure 1.

Response: We are grateful for this suggestion and a similar experiment is pursued using super resolution microscopy. We found a great colocalization between Tmem65 and β 1 in en-face orientation. The result is included in the revised manuscript (Fig. 7a).

Minor

Introduction Line 90-92: “ Conversely, Cx43 trafficking and localization to the ICD depends on desmosomes 12 and tight junction proteins 8 possibly through at specialized ICD region called the perinexus.”

Probably best not to invoke tight junctions - cardiomyocytes do not have these junctional structures

Response: We appreciate the reviewer’s comment. ‘Tight junction’ discussion has been removed from the text. (Page 4, lines 89-90).

Reviewer #3 (Remarks to the Author):

Teng et al. report their results on AAV-mediated expression of a sh-RNA for knockdown of Tmem65 in neonatal mice. Mice developed cardiac hypertrophy and conduction abnormalities 3 weeks after AAV-shRNA application and dilated cardiomyopathy 7 weeks after AAV injection. Mechanistically, Tmem65 knock-down resulted in loss of ephaptic junctions at the intercalated disc. Overall, the results are novel and help to understand the role of Tmem65 with intercalated discs. Nevertheless, a number of issues needs further clarification.

Major:

1. The potential role of the gender is an interesting finding. However, the role of the gender for the findings reported in this manuscript are not clear. Could gender have played a role in the mice having died after intramyocardial injections? Has gender been determined in these mice or control mice? Overall, considering a gender effect, it would be helpful to specify gender information in all figure legends and tables.

Response: We agree with the reviewer that sex based differences could be extremely interesting with differential responses to Tmem65 silencing in a gender-related manner, but the exact mechanism remains unknown. Speculation of different ideas has been proposed in the lab and one of these theories is included in the discussion (Pages 27, lines 567-571). Ongoing pursuit has been carried out in the lab and more convincing evidence will be needed for better determining the sex-specific response to the loss of cardiac Tmem65 in vivo. In our revision, we have ensured that all sex information is now included in all figures, legends, and tables as advised by the reviewer.

2. Animal numbers are not clearly indicated for all experiments. How many mice injected to investigate the early and late time point (figure 4, 5, and 6 or figure 2 and 3)? I did not find information on animal numbers (or gender) used for distinct measurements in the figure legends (only in the tables).

Response: This is an important feature we need to update. Animal numbers are now included in the all figure legends of the revised manuscript for clarification as suggested by the reviewer.

3. It is difficult to spot differences in Scn1beta immunofluorescence in in Fig. 6e.

Response: We thank the reviewer for their comment which we agree with. β 1 was recently found localized to the perinexus region of the intercalated discs in the heart (Veeraraghavan et al, 2018, Elife). Consistently, we show that both β 1 and the intercalated discs are usually perpendicular or position at an angle to myofibers. In comparison, β 1 in Tmem65 KD cardiomyocytes is longitudinally distributed on muscle fibers (Fig. 6e, white arrows) or form aggregates (Fig. 6e, yellow arrows) in Tmem65 KD hearts. The explanation is now incorporated in the revised manuscript for reader clarity (Pages 23-24, lines 498-501).

4. Analysis of fibrosis on the histological level (Masson Trichrome or Sirius red stains) would be

helpful.

Response: We thank the reviewer for the suggestion. Quantification of cardiac fibrosis is now included in the revised manuscript (Page 19, lines 401-407. Fig. 3e).

5. Please provide details on the shRNA sequence. Was it expressed within a microRNA context to be suitable for expression with the chicken troponin T promoter?

Response: The shRNA construct is indeed embedded in a microRNA-based format for an efficient conversion. The exact sequence is now included in the revised manuscript (Page 6, line 112).

Minor:

Line 46: Introduce „KD“ in the abstract.

Response: This is now included in the revised manuscript (Page 2, line 64).

Line 50: Delete the comma.

Response: Comma has been removed from the revised manuscript. (Page 2, line 68).

Line 102: grammar – rather „interact“?

Response: The revised manuscript now has the correct grammar. (Page 5, line 121).

Line 110: Provide reference.

Response: The reference has been provided accordingly. (Page 7, line 157).

Line 124: Please cite the correct source of pDG9.

Response: The source has been provided. (Page 6, line 127).

Figure 8: It is difficult to understand that the left part is the normal situation (scrambled shRNA) while the right part of the sketch is the Tmem65 KD situation. Please revise the otherwise helpful figure.

Response: Reflecting on this figure, we agree that we spent a great deal of time and effort in illustrating how Tmem65 KD negatively impacts the intercalated discs, but did not properly introduce the ‘normal’ situation (scrambled shRNA) in our first submission. We now include a better coverage of the model system in normal hearts in the revised manuscript for clarification (Pages 26, lines 548-552).

REVIEWER COMMENTS

Reviewer #1 (Remarks to the Author):

The authors have significantly improved the manuscript. I remain concerned, though, about the lack of evidence regarding basic electrophysiological parameters. I tried to make this point in the previous review, and in response the authors measured action potential morphology and calcium transients by optical methods. This, in my opinion, is far from sufficient.

I do realize that the authors are focused on the ephaptic coupling hypothesis. But I think that they do so while disregarding alternative (and quite likely) possibilities. While electric field-mediated transfer of charge is possible, one cannot ignore the fact that this phenomenon would be happening parallel to a low-resistance pathway (the gap junction channel), which is likely to shunt the current away. The authors provide evidence for Cx43 internalization. That is likely to reduce junctional conductance. Measurements of junctional conductance, and consideration of the fact that propagation may be impaired due to loss of this component, seem necessary.

I thank the authors for heeding my concern regarding the fact that sodium channels are also located in other territories of the cell. My point however is not only about their localization, but about their function. If sodium channels at the perinexus are critical for propagation, they should then generate enough current so that their absence can be detected by patch clamp methods, be it whole-cell, or macropatch. As the authors know very well, the abundance of a channel protein and its localization are not direct measurements of its function as a carrier of charge. Patch clamp is. This method is generally available and without it, in my opinion, this study is incomplete. If the authors are assessing the mechanism for reduced conduction velocity, then a direct method for evaluating sodium current seems a must.

In this version, the authors show changes in electrically-evoked calcium transients. Surprisingly, they do not measure calcium currents, thus ignoring one half of the calcium induced-calcium release process. The observed changes in calcium transients may be signaling that calcium currents have been affected. The latter would be particularly important in the context of the cardiomyopathic phenotype.

Can one have altered calcium currents and yet unaffected action potential morphology? Yes. If inward and outward currents both change, the net current across the membrane can be the same. Importantly, even if the morphology of the action potential is the same, the consequence, particularly in the case of a cardiomyopathic heart, can be different. The new data provided (altered calcium transients and yet unaffected action potential duration) begs for an analysis of outward currents.

In summary, I believe that this paper would be greatly improved if the authors were to include measurements of gap junctional conductance, sodium current (ideally in a region-specific manner), calcium currents and outward currents. I also believe that without those measurements, the authors fall short of providing enough evidence to fully substantiate their conclusions.

Reviewer #3 (Remarks to the Author):

The manuscript has improved in clarity. Nevertheless, there are a couple of open issues:

Although there are sex differences in survival, data from male and female mice are still

presented together. Sex of mice and animal numbers have been now provided in figure legends, but tables 2 and 3 still lack information on sex. However, considering the low standard deviation, sex differences might be low. Nevertheless, suppl. data comparing effects in males and females separately might still be helpful.

page 27, line 573 Why "invaluable tool"? This appears contradictory.

page 36, line 793 should be "scrambled" instead of "scram"

page 36, line 802: "Experiments were performed in mice of both sexes, 4 mice per groups." Does this refer to the complete figure 6? How about controls (n=10)?

Reviewer #4 (Remarks to the Author):

The authors have been appropriately responsive to the critiques of the reviewers. The addition of new data has added important rigor as well as mechanistic insight to the study.

Reviewer #1 (Remarks to the Author):

The authors have significantly improved the manuscript. I remain concerned, though, about the lack of evidence regarding basic electrophysiological parameters. I tried to make this point in the previous review, and in response the authors measured action potential morphology and calcium transients by optical methods. This, in my opinion, is far from sufficient.

#1: I do realize that the authors are focused on the ephaptic coupling hypothesis. But I think that they do so while disregarding alternative (and quite likely) possibilities. While electric field-mediated transfer of charge is possible, one cannot ignore the fact that this phenomenon would be happening parallel to a low-resistance pathway (the gap junction channel), which is likely to shunt the current away. The authors provide evidence for Cx43 internalization. That is likely to reduce junctional conductance. Measurements of junctional conductance, and consideration of the fact that propagation may be impaired due to loss of this component, seem necessary.

Response to Reviewer

We thank the reviewer for this comment and appreciate the concern regarding the direction of our work. The reviewer is correct that we observed Cx43 internalization in the adult heart sections and it would be reasonable to conclude if the junctional conductance would be affected under these conditions. In fact, we did perform those experiments previously using viral knockdown of Tmem65 in mouse neonatal myocytes, which showed significantly reduced junctional conductance in the Tmem65 shRNA transduced cells (PMID 26403541). These experiments were performed using dye injection tracking studies and microelectrode array (MEA) plate studies.

In the spirit of providing more clarity on the objectives, results, data interpretation and discussion, we have undertaken a major rewriting of the paper, particularly the Results and Discussion sections. At the request of the reviewer, we have been considering alternative possibilities, beyond alterations in junctional conductance via the changes in perinexal nanodomains. In so doing, we have included new whole heart optical mapping studies which definitely establish reduced electrical conduction in the Tmem65 KD hearts (Pages 27-28, lines 599-609, Figs 2g and 2i, Table 3). We also performed whole cell patch-clamp studies on isolated adult myocytes that were also requested by the reviewers which showed no change in the magnitude of the whole-cell voltage-gated Na⁺ currents (I_{Na}). While these results were initially unexpected, they are in fact consistent with several previous studies, which have concluded that channels at the intercalated discs (including Nav1.5 channels and Cx43 hemichannels) cannot be measured using whole-cell patch-clamp recordings (Please see our response to the next point). Accordingly, in the revised manuscript we now consider carefully the impact of Tmem65 KD on electrical conduction as a result of Cx43 internalization as well as the reduced Nav1.5 channel numbers at the perinexus (Pages 38-40, lines 819-877) and we discuss these factors and their inter-dependence with fibrosis and changes in fundamental cellular architectures on the electrical changes induced by Tmem65 KD (Pages 43-44, lines 941-953). We also discuss the limitations of these studies as they must be performed

whereby the cell–cell connections are destroyed through the enzymatic dissociation of the myocytes (Page 39, lines 832-844).

#2. I thank the authors for heeding my concern regarding the fact that sodium channels are also located in other territories of the cell. My point however is not only about their localization, but about their function. If sodium channels at the perinexus are critical for propagation, they should then generate enough current so that their absence can be detected by patch clamp methods, be it whole-cell, or macropatch. As the authors know very well, the abundance of a channel protein and its localization are not direct measurements of its function as a carrier of charge. Patch clamp is. This method is generally available and without it, in my opinion, this study is incomplete. If the authors are assessing the mechanism for reduced conduction velocity, then a direct method for evaluating sodium current seems a must.

Response to Reviewer

Again, this is a relevant concern raised here. As the Reviewer correctly points out, NaV channels are not only found at the ICD regions of cardiomyocytes. As already mentioned, we performed new whole-cell patch-clamp experiments to measure directly Na⁺ currents, as well as K⁺ and Ca²⁺ currents (to address other comments by the Reviewers), in isolated cardiomyocytes.

To the Reviewer's point, our patch clamp measurements of I_{Na} revealed no significant differences between scrambled (control) and Tmem65 KD mice groups. While this data would support the reviewers concern and suggest that sodium channels at the perinexus might not be critical for propagation, there is another interpretation of these results that is anticipated and consistent with previous studies. Specifically, during the process of CM isolation, the function of these channels is disrupted. Indeed, in the first/early studies by Cleeman and Morad describing heart digestions, it was shown by EM that many isolated cells had pieces of membrane (membrane loops) covering the ICD regions (PMID 3971501; 7143548); the idea proposed by these authors was that this was a requirement for CM survival by essentially covering the gap junctional channels (GJCs) and the intercalated disc regions. We now know that GJCs and hemichannels of connexins close under harsh conditions of high Ca²⁺ and low pH, so it may not be necessary to have membrane covers overtop the ICD. This may explain why several previous studies have established that the regions of intercalated discs undergo degeneration after cell isolation (PMID 3971501, 7143548). This interpretation of our I_{Na} results is consistent with previous work by co-authors on this paper which showed that when the perinexus structure is disrupted and Nav1.5 channels are lost (PMID 30106376) there is no change in the whole-cell currents (Figs. 4d-g). To summarize, as requested we measured directly electrical conduction along with membrane currents in our model. The I_{Na} results do not allow us to make definitive unequivocal conclusions regarding the potential role of I_{Na} changes in slowed cardiac conduction when Tmem65 levels are reduced. Nevertheless, the findings provide further evidence supporting the complications and challenges associated with assessing channel function at the intercalated discs in isolated cardiomyocytes using the

whole-cell patch-clamp technique; this point is now made explicitly in our discussion. We feel it is also worth mentioning that I_{Na} in CMs from Tmem65 KD hearts showed far greater variability in peak I_{Na} compared to the scrambled (control) mice. This difference in variability could arise from many factors such as differences in regional heterogeneity or possibly the variations in impact of cell isolation on intercalated disc region.

To overcome the limitations of assessing I_{Na} using whole-cell patch-clamp in isolated cardiomyocytes, it will be necessary to consider alternative approaches such as combining smart patch clamp technology using high-resolution imaging (as in PMID 30106376). Based on our I_{Na} measurements, we have reoriented the Discussion of our paper to acknowledge the limitations of our measurements in making definitive conclusions linking change in I_{Na} to the phenotypic changes in these mice. Parenthetically, we include in our revised Discussion the potential consequences of altered heart function arising from the cardiomyopathy on the electrical measurements (Pages 38-42, line 813-902).

#3. In this version, the authors show changes in electrically-evoked calcium transients. Surprisingly, they do not measure calcium currents, thus ignoring one half of the calcium induced-calcium release process. The observed changes in calcium transients may be signaling that calcium currents have been affected. The latter would be particularly important in the context of the cardiomyopathic phenotype.

Response to Reviewer

Similar to the above points, we have now performed whole-cell patch clamp measurements, including calcium currents through voltage-gated calcium channels. Importantly, we found that peak I_{Ca} current density was reduced in Tmem65 KD group compared to control mice (-11.33 ± 0.24 mV versus -13.82 ± 0.32 mV, $p < 0.01$), suggesting reduced I_{Ca} may contribute to decreased calcium transient amplitudes by decreasing the trigger calcium release from the sarcoplasmic reticulum (i.e., reduced EC coupling gain). Tmem65 KD also caused very small changes in the activation slopes (5.83 ± 0.28 in the control versus 6.86 ± 0.22 in the knockdown group, $p < 0.05$) and the $V_{1/2}$ of inactivation of calcium channel (-27.75 ± 0.39 mV in control versus -23.88 ± 0.18 mV in Tmem65 KD, $p < 0.01$). While the relevance of this observation to electrical conduction is unclear in the ventricle, since I_{Ca} contributes little minimally to ventricular conduction, this could help explain the prolonged PR intervals since I_{Ca} is crucial for (slow) conduction through the AV node. The basis for reduced I_{Ca} in the Tmem65 KD group is unclear. However, this could arise from the cardiomyopathic phenotype in these mice, as shown in other studies (PMID: 32580140). Clearly further studies will be needed to assess the direct effects of Tmem65 knockout on I_{Ca} versus indirect effects arising from the induced heart disease (PMIDs: 26577135 and 12554098).

#4. Can one have altered calcium currents and yet unaffected action potential morphology? Yes. If inward and outward currents both change, the net current across the membrane can be the same. Importantly, even if the morphology of the action potential is the same, the consequence, particularly in the case of a cardiomyopathic heart, can be different. The new data provided (altered calcium transients and yet unaffected action potential duration) begs for an analysis of outward currents.

Response to Reviewer

This is also a good and important point which becomes even more cogent given our I_{Ca} measurements. The I_{Ca} data shows that consistent with Ca^{2+} transient disruption and reduced contractile performance after Tmem65 KD, we observed reduced I_{Ca} . As the Reviewer correctly points out, this is expected to impact AP profiles. To better understand the impact of these many changes on AP profiles, we also quantified the various outward voltage-dependent potassium currents (Kv), using our previously published robust fitting approaches (PMID: 21253754 Liu et al, BRC). These results revealed that Tmem65 reductions lead to extremely small increases the fast transient outward current ($I_{to,f}$) with somewhat larger increases in the slow transient outward K^+ current ($I_{to,s}$). Since the contribution of $I_{to,s}$ to AP durations remains unclear because of its slow recovery from inactivation, these results suggest that changes in transient outward K^+ current associated with Tmem65 reductions contribute minimally to AP duration. On the other hand, $I_{K,slow}$ is markedly reduced in the Tmem65 KD CMs without changes in the other major repolarizing current, $I_{K,slow2}$. Since $I_{K,slow1}$ contributes very importantly to AP profile/duration, this readily explains why AP are not markedly changed despite the reductions in I_{Ca} . The implications of our K^+ current measurements and dissection are discussed on Pages 41-42, lines 878-902.

It should be mentioned that the very small increases in $I_{to,f}$ in Tmem65 KD hearts is somewhat unexpected since this current is typically reduced quite profoundly in most heart disease conditions (see review PMID: 25646587). In this regard, the dilated cardiomyopathic phenotype induced by Tmem65 KD may reflect the short-term nature of studies as well as the use of very young mice, which may not recapitulate the known changes in ion channel expression and activity with chronic disease. Indeed, a previous study showed that in a mouse model of DCM, heart enlargement and only minimal changes in APD were observed at 1-month, with significant downregulation of Kv4.2, Kv1.5 and KCHIP2 mRNA and protein levels, APD prolongation, and increased mortality occurring between months 1-3 of disease (PMID: 22514734). This study would be consistent with there only being modest electrical remodeling at the time point (3-weeks) we assessed given the age- and sex-dependence of mortality rates in our model.

#5. In summary, I believe that this paper would be greatly improved if the authors were to include measurements of gap junctional conductance, sodium current (ideally in a region-specific manner), calcium currents and outward currents. I also believe that without those measurements, the authors fall short of providing enough evidence to fully substantiate their conclusions.

Response to Reviewer

Consistent with all of the earlier comments, we are grateful to the reviewer for ensuring that we completed the necessary and appropriate studies. The results for both optical mapping and patch clamp experiments are now included in the revised manuscript (Figs 2g and 2i, Table 3; Fig 6, Table 5).

Reviewer #3 (Remarks to the Author):

The manuscript has improved in clarity. Nevertheless, there are a couple of open issues:

#1. Although there are sex differences in survival, data from male and female mice are still presented together. Sex of mice and animal numbers have been now provided in figure legends, but tables 2 and 3 still lack information on sex. However, considering the low standard deviation, sex differences might be low. Nevertheless, suppl. data comparing effects in males and females separately might still be helpful.

Response to Reviewer

Thank you. We have updated the manuscript to include information in the figure captions that highlight the number of mice from each sex that were used in the measurements for both all Figure legends and Tables (Page 45-60). The Reviewer is correct to mention that given the low standard deviations in many of our outcome measurements that we believe that sex differences in the individual measurements appears low, with the study currently statistically underpowered to be able to sufficiently address the question of sex differences in mortality rates between male and female mice in response to Tmem65 knockout. Nevertheless, we now include some hypotheses and literature support in the Discussion for speculating sex differences in animals (Pages 46-51, lines 980-1190).

page 27, line 573 Why "invaluable tool"? This appears contradictory.

Response to Reviewer

We thank reviewer's comment and have modified the sentence accordingly (Page 36, lines 783-787).

page 36, line 793 should be "scrambled" instead of "scram"

Response to Reviewer

We have modified the text for a better clarity (Page 36, lines 783-787)

page 36, line 802: "Experiments were performed in mice of both sexes, 4 mice per groups." Does this refer to the complete figure 6? How about controls (n=10)?

Response to Reviewer

We thank reviewer's comments. A more comprehensive experiment was performed per request of other reviewers for measuring both conduction velocity and action potential in the heart and isolated adult cardiomyocytes. The new data have been added for replacing action potential experiments in the last version of the manuscript. The number of cardiac samples and the genders of animals are reported in Figures legends and Table legends (Page 46-51).

Reviewer #4 (Remarks to the Author):

The authors have been appropriately responsive to the critiques of the reviewers. The addition of new data has added important rigor as well as mechanistic insight to the study.

REVIEWERS' COMMENTS

Reviewer #1 (Remarks to the Author):

I thank the authors for conducting extensive experimental studies to address my concerns. I have no further comments.

Reviewer #3 (Remarks to the Author):

Overall, the manuscript has further improved by pointing out differences in survival between male and female mice and further electrophysiological data. Nevertheless, the issue of different sexes of mice used in the investigation is still not fully resolved. It is not clear to me whether the Kaplan-Meier curves in fig. 1 display only male mice. Furthermore, Suppl. fig. 2 lacks information about sex.

Reviewer #1 (Remarks to the Author):

I thank the authors for conducting extensive experimental studies to address my concerns. I have no further comments.

We thank the Reviewer for insightful comments.

Reviewer #3 (Remarks to the Author):

Overall, the manuscript has further improved by pointing out differences in survival between male and female mice and further electrophysiological data. Nevertheless, the issue of different sexes of mice used in the investigation is still not fully resolved. It is not clear to me whether the Kaplan-Meier curves in fig. 1 display only male mice. Furthermore, Suppl. fig. 2 lacks information about sex.

The sexes and numbers of animals used for generating the Kaplan-Meier plot in Fig. 1b (Page 47, lines 996-999) and Supp. Fig 2 have now been included in the text (Page 51, lines 1152-1154). In both incidents, both male and female mice were included to reflect the survival and ECG measurements in tested populations. We are grateful for Reviewer's attention to details, which help to refine the manuscript.